# USP7/Maged1-mediated H2A monoubiquitination in the paraventricular thalamus: an epigenetic mechanism involved in cocaine use disorder

Julian Cheron [1], Leonardo Beccari [2,3], Perrine Hagué[1], Romain Icick[4], Chloé Despontin[1], Teresa Carusone [5], Matthieu Defrance[6], Sagar Bhogaraju [5], Elena Martin-Garcia [7,8,9], Roberto Capellan [7,8], Rafael Maldonado [7,8], Florence Vorspan[4], Jérôme Bonnefont [10] & Alban de Kerchove d'Exaerde [1,11] ✉

The risk of developing drug addiction is strongly influenced by the epigenetic landscape and chromatin remodeling. While histone modifications such as methylation and acetylation have been studied in the ventral tegmental area and nucleus accumbens (NAc), the role of H2A monoubiquitination remains unknown. Our investigations, initially focused on the scaffold protein melanoma-associated antigen D1 (Maged1), reveal that H2A monoubiquitination in the paraventricular thalamus (PVT) significantly contributes to cocaine-adaptive behaviors and transcriptional repression induced by cocaine. Chronic cocaine use increases H2A monoubiquitination, regulated by Maged1 and its partner USP7. Accordingly, Maged1 specific inactivation in thalamic Vglut2 neurons, or USP7 inhibition, blocks cocaine-evoked H2A monoubiquitination and cocaine locomotor sensitization. Additionally, genetic variations in MAGED1 and USP7 are linked to altered susceptibility to cocaine addiction and cocaine-associated symptoms in humans. These findings unveil an epigenetic modification in a non-canonical reward pathway of the brain and a potent marker of epigenetic risk factors for drug addiction in humans.

Drug addiction, defined as uncontrollable drug intake despite harmful consequences, is a major health problem[1,2]. The number of deaths associated with cocaine use disorder (CUD) is increasing, and in contrast to tobacco and opioid addiction, no medication has been approved for CUD[1]. Identifying risk factors and therapeutic targets is thus a pressing biomedical challenge. The neurobiological mechanisms that drive and sustain drug addiction involve the combination of neuronal and synaptic plasticity with modifications of gene expression, achieved, in part, through epigenetic mechanisms[3]. By regulating chromatin-related processes, drug-induced epigenetic alterations

[1]Université Libre de Bruxelles (ULB), ULB Neuroscience Institute, Neurophysiology Laboratory, Brussels, Belgium. [2]Centro de Biología Molecular Severo Ochoa, CSIC-UAM, Madrid, Spain. [3]Université Claude Bernard Lyon 1, Pathophysiology and Genetics of Neuron and Muscle, CNRS UMR 5261, INSERM U1315, Lyon, France. [4]INSERM UMRS_1144, Université Paris Cité, Paris, France. [5]European Molecular Biology Laboratory, Grenoble, France. [6]Interuniversity Institute of Bioinformatics in Brussels, Université Libre de Bruxelles (ULB), Brussels, Belgium. [7]Laboratory of Neuropharmacology-Neurophar, Department of Medicine and Life Sciences, Universitat Pompeu Fabra (UPF), Barcelona, Spain. [8]Hospital del Mar Medical Research Institute (IMIM), Barcelona, Spain. [9]Departament de Psicobiologia i Metodologia de les Ciències de la Salut, Universitat Autònoma de Barcelona, Cerdanyola del Vallès, 08193 Barcelona, Spain. [10]Université Libre de Bruxelles (ULB), ULB Neuroscience Institute, Institut de Recherches en Biologie Humaine et Moléculaire (IRIBHM), Brussels, Belgium. [11]WELBIO, Wavre, Belgium. ✉e-mail: adekerch@ulb.ac.be

contribute to the aberrant gene expression and cellular function that underlie drug addiction pathogenesis[4]. Histone post-translational modifications (PTMs) that control chromatin structures regulate drug-adaptive behaviors and the subsequent transition to addiction[4,5]. Several chromatin modifications in the ventral tegmental area and nucleus accumbens (NAc), including histone H3 methylation, H3 and H4 acetylation, have been implicated in drug addiction. Still, the contribution of other histones and their PTMs is unclear[4,6–9]. The promoter of melanoma-associated antigen D1 (Maged1) was identified as one of the targets of histone H3 acetylation, a canonical PTM, induced in the NAc by chronic cocaine treatment[9]. Subsequently, we identified Maged1 as a master regulator of cocaine reward and reinforcement[10]. Indeed, Maged1 KO mice display, in parallel to an abolished locomotor sensitization, altered cocaine-induced conditioned place preference (CPP) and cocaine self-administration[10]. To date, however, the molecular mechanisms involving Maged1 in these cocaine-related phenotypes have not been characterized[10–12]. Despite its ubiquitous expression in the brain and its essential role in cocaine-related behaviors, Maged1 is not required for cocaine-induced behaviors in dopaminergic or GABAergic cells[10], leaving excitatory glutamatergic cells as the main candidate. In this work, we demonstrate that chronic cocaine administration increases H2A monoubiquitination in several brain regions, including the NAc and a non-canonical reward region—the thalamus. Maged1 and the deubiquitinase USP7 are further identified as key regulators of this histone PTM within the paraventricular thalamus (PVT). Our findings underscore the significant role of H2A monoubiquitination in driving cocaine-adaptive behaviors and transcriptional repression triggered by cocaine exposure. Furthermore, we uncover genetic variations in MAGED1 and USP7 associated with altered susceptibility to cocaine addiction and to cocaine-induced aggressive behavior in human subjects.

## Results

### Maged1 expression in Vglut2 neurons, and specifically in the paraventricular thalamus, is necessary and sufficient for cocaine-related behaviors

Here, we specifically inactivated Maged1 in glutamatergic Vglut1 or Vglut2 cells by crossing Vglut1- or Vglut2-Cre-driver mice with Maged1 floxed mice (Maged1[loxP]). Notably, only Vglut2-mediated (but not Vglut1-mediated) Maged1 inactivation (Maged1 conditional knockout (cKO)) abolished cocaine-induced locomotor sensitization (20 mg/kg), a straightforward behavioral paradigm used to model drug-adaptive behavior[13–15] (Fig. 1 and Supplementary Fig. 1). In rodents, repeated cocaine injections induce gradually increased locomotor activity; classically after five days of consecutive injections, the locomotor response reaches a ceiling level. This state lasts for months after cocaine withdrawal[13]. Locomotor sensitization is thus thought to be linked with important aspects of vulnerability to drug addiction[16,17], relapse, and drug craving[18,19]. We further excluded the contribution of glutamatergic telencephalic neurons by using an Emx1-Cre driver line (Supplementary Fig. 2). Indeed, specific inactivation of Maged1 in telencephalic Emx1 cells did not alter cocaine-induced locomotor sensitization (Supplementary Fig. 2). Further, to delve into Maged1's role in the reinforcing properties of cocaine within Vglut2 neurons, we conducted a CPP experiment and operant self-administration assessments (Supplementary Fig. 3). Our CPP results demonstrated a significant distinction between the two groups. Specifically, Vglut2-mediated Maged1 inactivation reduced time spent in the paired compartment associated with cocaine (Supplementary Fig. 3a). This finding suggests that Maged1 has a substantial influence on the response to cocaine-associated stimuli and the development of related adaptive behaviors. Shifting our focus to operant self-administration assessments (Supplementary Fig. 3b–d), we observed that Maged1 inactivation in Vglut2 neurons did not lead to alterations in learning during the training sessions (fixed ratio 1 and 3, Supplementary Fig. 3b, c). However,

intriguingly, it did lead to a significant increase in the count of active nose-poking responses during the three consecutive days following learning (refer to supplementary results and Supplementary Fig. 3d). This outcome further supports the notion of Maged1 early involvement in early cocaine adaptive behaviors.

To screen for potential Vglut2 nuclei involved in the acquisition of this phenotype, we performed adeno-associated virus (AAV) retro mCherry injection in the NAc and revealed the amygdala, the ventral subiculum, and the PVT as the main Vglut2 nuclei targeting the NAc (Fig. 2a, b). Of these, Maged1 mRNA in situ hybridization and immunohistochemistry (IHC) showed a singularly high level of Maged1 expression (Fig. 2c) in the PVT, which has recently been shown to be involved in drug addiction[20–23]. We therefore examined the impact of Maged1 inactivation in the PVT via stereotactic injection of an AAV encoding a Cre recombinase in adult Maged1[loxP] mice (Fig. 2d, e). Here again, we observed the abolition of cocaine-induced locomotor sensitization; thus, phenocopying Maged1-KO mice and excluding developmental effects (Fig. 2f). To test the sufficiency of Maged1 in the PVT to cocaine sensitization, we stereotactically injected an AAV expressing Maged1 into Maged1-KO mice and found that cocaine-evoked sensitization was restored (Fig. 2d, e, g). In contrast, stereotactic inactivation[10] (Supplementary Fig. 4) or re-expression (Supplementary Fig. 5) of Maged1 in other Vglut2- or mixed Vglut1/Vglut2 cell hubs of drug addiction, such as the prefrontal cortex (PFC), amygdala or ventral subiculum[24], did not lead to major significant changes in cocaine-induced locomotor sensitization. Taken together, these results point toward PVT as the key nucleus where Maged1 plays a role in drug addiction.

### Cocaine-induced transcriptomic changes involve the Maged1-mediated Polycomb repressive complex

To identify the molecular mechanisms linking Maged1 in the PVT to cocaine-mediated behavior, we analyzed the transcriptome in the thalamus after chronic cocaine (or saline) administration in control and Maged1-cKO mice by RNA sequencing (RNA-seq) (Fig. 3a). In control mice, repeated cocaine injections induced an upregulated expression of 252 genes and a downregulated expression of 452 genes compared to the expression of these genes after saline treatment (Fig. 3b). The transcriptional response (and particularly the downregulation process, see supplementary results) to cocaine exposure by Maged1-cKO mice differed from that of control mice in terms of genes and their assigned GO terms (Fig. 3b and Supplementary Fig. 6 and 7a, and Supplementary Data 1). Interestingly, a gene set enrichment analysis with curated gene sets revealed that cocaine-regulated genes in control but not in Maged1-cKO mice were targets of the Polycomb repressive complex (PRC) (Fig. 3c, d and Supplementary Data 1). Briefly, PRC2 mono-, di-, and tri-methylates lysine 27 of histone H3 (H3K27me1/me2/me3)[25,26], and PRC1 monoubiquitinates histone H2A on lysine 119 (H2AK119ub1)[27] (Fig. 3d). Notably, the genes with expression similarly regulated by cocaine in the control and Maged1-cKO mice were not known to be targeted by PRC action, as we did not find any significant overlap between genes known to be targeted by PRC and these Maged1-independent genes.

### Maged1 enables interaction between histone H2A and USP7

To understand how Maged1 mediates the profound transcriptional changes observed upon cocaine exposure (Fig. 3b–d), we aimed to identify Maged1 protein partners in the thalamus by performing co-immunoprecipitation and mass spectrometry (co-IP-MS) experiments with a Maged1 antibody that recognizes both the full length and truncated form (functional KO) in saline- and cocaine-treated mice (Fig. 3e). We identified 404 potential Maged1-interacting proteins (Supplementary Results and Supplementary Data 2). Among them, histone H2A, in its abundantly expressed H2A.2 variant in the adult brain[28,29], and de-ubiquitinase USP7, a well-characterized PRC1

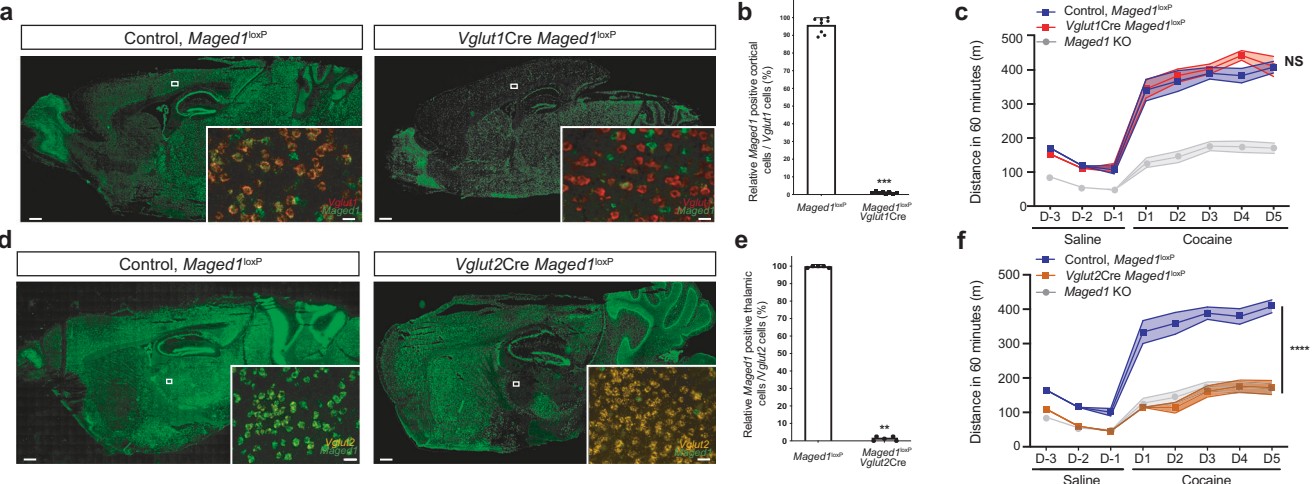

**Fig. 1 | *Maged1* inactivation in Vglut2 neurons completely recapitulates the effect of *Maged1* global inactivation.** Fluorescent in situ hybridization (FISH) of *Maged1* mRNA (in green) (scale bar: 500 μm). Insets show multiplexed FISH with *Vglut1* (in red) (**a**), *Vglut2* (in orange) (**d**), and *Maged1* mRNA (×20 magnification) in the cortex (**a**) and in the thalamus (**d**) (scale bar: 20 μm) (**a**), 30 μm (**d**). Relative percentage of *Maged1* mRNA-positive cells among *Vglut1* mRNA-positive cells in the cortex, P = 0.0002 (**b**), and *Vglut2* mRNA-positive cells in the thalamus, P = 0.0079 (**e**). n = 7 for each group (Mann–Whitney test (two-tailed). **c, f** Cocaine-induced

locomotor sensitization, 20 mg/kg, intraperitoneal (i.p.) injection, Maged1 KO, n = 15. In panel (**c**): control, *Maged1^loxP^*, n = 13; *Vglut1*cre, *Maged1^loxP^* mice, n = 7, (two-way ANOVA, control versus *Vglut1*Cre::*Maged1^loxP^*, P = 0.6651; days, P < 0.0001; interaction factor (genotype X days) P = 0.5056). In panel (**f**): control, *Maged1^loxP^*, n = 12; *Vglut2*Cre::*Maged1^loxP^* mice, n = 9, (two-way ANOVA, control versus *Vglut2*Cre::*Maged1^loxP^*, P < 0.0001; days, P < 0.0001; interaction factor (genotype x days), P < 0.0001). Source data are provided as a Source Data file.

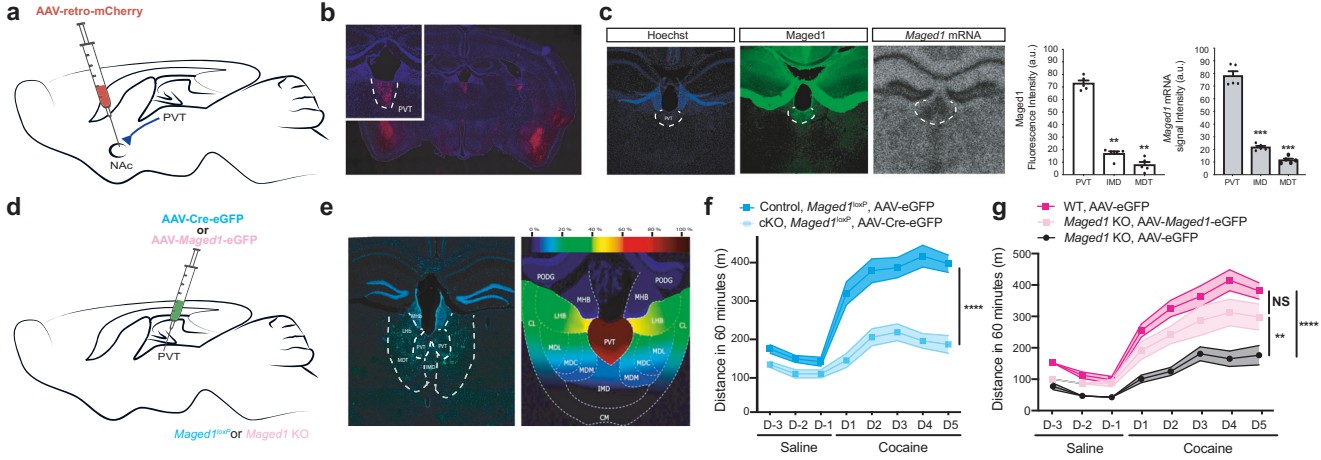

**Fig. 2 | *Maged1* is 'necessary and sufficient' in the PVT. a** Schematic of adeno-associated virus (AAV)-retro-mCherry injection in the nucleus accumbens (NAc). **b** Wide-field image showing the mCherry signal specifically in the PVT and other NAc-projecting regions. Inset shows mCherry signal in the PVT. **c** Immunohistochemistry (IHC) staining of Maged1 (in green) and Hoechst (blue) and in situ hybridization autoradiogram of *Maged1* mRNA and their quantification n = 5 mice (for Maged1 immunofluorescence: Repeated-measure one-way ANOVA (***P = 0,0002) followed by Tukey's multiple comparisons tests (PVT vs IMD, P = 0.0027; PVT vs MDT, P = 0.002; IMD vs MDT P = 0.1552); for *Maged1* mRNA: Repeated-measures one-way ANOVA (P < 0.0001) followed by Tukey's multiple comparisons post hoc test (PVT vs IMD, P = 0.0007; PVT vs MDT, P = 0.0002; IMD vs MDT P = 0.0734). **d** Schematic of inactivation (AAV-Cre-eGFP into *Maged^loxP^* mice) and re-expression experiments in the PVT (AAV-*Maged1*-eGFP in the *Maged1*-KO mice). **e** IHC staining showing the eGFP signal (in green) in the PVT (-2 mm from the bregma) (left), topographic representation of the targeted area; colors

represent the percentage of the superimposed eGFP-targeted area (right). Inactivation experiment (**f**) and re-expression experiment (**g**) consisting of a cocaine-induced locomotor sensitization, 20 mg/kg, ip injection. In panel (**f**): control, *Maged1^loxP^*, AAV-eGFP mice, n = 10; cKO, *Maged1^loxP^*, AAV-Cre-eGFP mice, n = 11, (two-way ANOVA, control versus *Maged1*-cKO, P < 0.0001; days, P < 0.0001; interaction factor (genotype x days) P < 0.0001). In panel (**g**): control *Maged1*-KO, AAV-eGFP mice, n = 10; control *Maged1* WT, AAV-eGFP mice, n = 10; re-expressed in KO, AAV-*Maged1*-eGFP mice, n = 8, (two-way ANOVA, groups P < 0.0001; days, P < 0.0001; interaction factor (genotype X days) P < 0.0001 and Sidak's test; re-expressed in KO versus control *Maged1* WT mice, P = 0.0731; re-expressed in KO versus control Maged1-KO mice, P = 0.0072; control *Maged1* WT versus control *Maged1*-KO, P < 0.0001). Data represented as mean±s.e.m. PVT: paraventricular nucleus of the thalamus, MDT: mediodorsal thalamic nucleus (including the medial, lateral, and central parts), IMD: intermediodorsal nucleus of the thalamus. Source data are provided as a Source Data file.

component[30–32], as well as several other proteins indirectly linked to PRC complexes, such as Praja1, hnRNPA2/B1, Myo5a, Dbn1 and Neb1/2, were identified (Fig. 3f)[33,34]. Within all the identified candidates, histone H2A was the only one whose (i) interaction with Maged1 was

disrupted in the *Maged1*-cKO samples and (ii) displayed a differential association with Maged1 in cocaine and saline-treated control mice (Fig. 3f and Supplementary Data 2). USP7 behaved as a Maged1 partner under both saline and cocaine treatment conditions (Fig. 3f

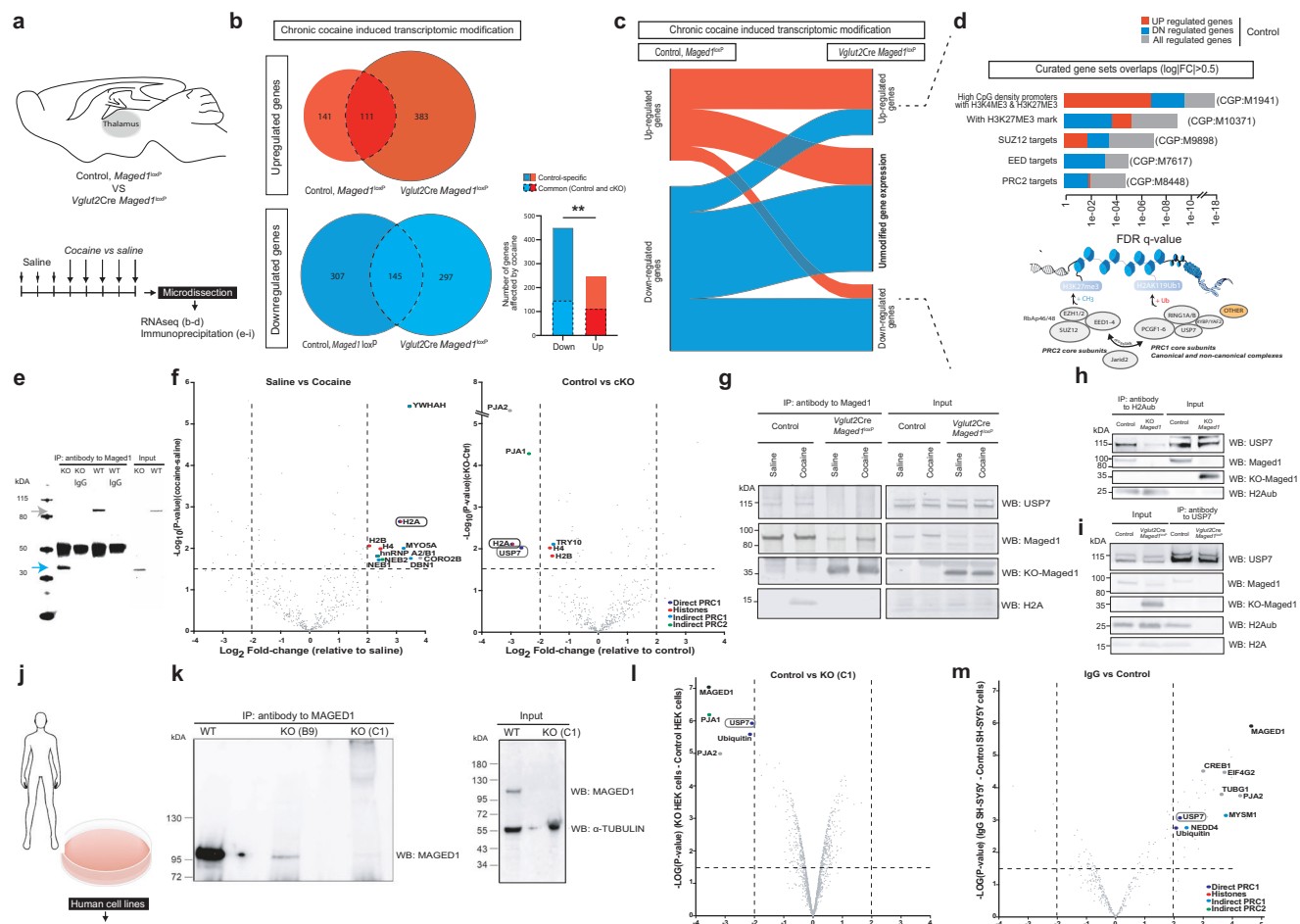

**Fig. 3 | *Maged1* inactivation modifies gene expression regulated by the Polycomb pathway through the USP7-H2A interaction. a** Mouse Thalamic RNA Sequencing and Immunoprecipitation (IP) Analysis. **b** Venn diagram showing differentially regulated genes in control mice, *Magel1loxP* mice and *Maged1*-cKO, *Vglut2*Cre *Maged1loxP* mice (log2|FC | ≥ 0.5, FDR < 0.05). Proportions of overlapping genes between the control and *Maged1*-cKO group are indicated (chi-square test, two-sided, **P = 0.0016). **c** Alluvial plot depicting gene expression changes in the control and their potential modifications in *Maged1*-cKO mice (log2|FC | ≥ 0.5, FDR < 0.05). **d** Curated gene set analysis showing overlapping between literature-based gene categories and unmodified genes in *Maged1*-cKO mice (top) and scheme of the Polycomb repressive complexes, PRC1 and PRC2 (bottom). IgG or anti-Maged1 antibody-based thalamic IP followed by western blotting (WB) (**e**) or IP-Mass spectrometry (MS) (**f**) for Maged1; WT band is indicated with a gray arrow, truncated protein (functional KO) is indicated with a blue arrow. **f** Mass spectrometry (MS) results for protein abundance between control mice treated with saline (n = 5) and control mice treated with cocaine (n = 7) (left side), and between control

mice (n = 7) and *Maged1*-cKO, *Vglut2*Cre::*Maged1loxP* (n = 8) mice (both treated with cocaine) (right side). **g** Mononucleosome (Mn) IP of Maged1 with USP7 and H2A identification by WB. **h** MnIP of monoubiquitinated H2AK119 (H2Aub) with identification of Maged1 and USP7 by WB. **i** MnIP of USP7 with identification of Maged1 and H2Aub and H2A by WB. Panels (**h, i**) were performed under saline treatment conditions. **j** Human cell IP method (panels **k, l, m**). **k** WB analysis of HEK293T lysates for MAGED1 in wild type (WT) and C1 subclones (right) and IP of endogenous MAGED1 in WT, B9 and C1 subclones (left). Volcano plots showing MS results for protein abundance in MAGED1 IP from WT versus C1-KO HEK293T cells (**l**) and in the WT versus IgG in SH-SY5Y cells (**m**). For MS results (**f, l, m**), proteins meeting the specified threshold (MaxQuant algorithm FDR = 1 % on peptide-to-spectrum matches, -log10 P value > 1.5 and log2|FC | > 2) are highlighted with a color code (dark blue for direct PRC1 component, red for histones, blue for known PRC1 interactors, and green for known PRC2 interactors). Source data are provided as a Source Data file.

and Supplementary Data 2). To confirm these interactions, we performed Maged1 co-IP with mononucleosome samples extracted from the thalamus, followed by western blotting (WB) for protein identification. We confirmed that the specific link between Maged1 and H2A was evident only in the presence of cocaine, while USP7 coprecipitated with Maged1 under both cocaine and saline treatment conditions (Fig. 3g). In agreement with these IP data, we observed that Maged1 and USP7 protein levels were both higher in the PVT than in other parts of the thalamus, like the adjacent mediodorsal thalamic nucleus (MDT) (including its medial, lateral and central parts) and the intermediodorsal nucleus of thalamus (IMD). (Supplementary Fig. 8).

Next, we postulated that the Maged1 interaction with H2A and USP7 contributes to the regulation of the monoubiquitination of H2A

at lysine 119 (H2AK119ub1 (hereafter H2Aub)) as Mage proteins assemble with different E2/3 ubiquitin ligases[12,35]. This histone mark deposited by PRC1 and linked to gene repression[36,37] may explain how Maged1 represses gene expression in thalamic neurons upon chronic cocaine administration. To test this hypothesis, we performed co-IP experiments using an anti-H2Aub monoclonal antibody and observed that USP7 and Maged1 both linked with H2Aub under saline and cocaine treatment conditions (Fig. 3h and Supplementary Fig. 9). Notably, USP7 binding to H2Aub was significantly decreased in the *Maged1*-KO mice, indicating that Maged1 was necessary for this H2Aub-USP7 interaction (Fig. 3h and Supplementary Fig. 9). Furthermore, USP7-specific antibody co-immunoprecipitated H2Aub and H2A under saline and cocaine treatment conditions but not in the *Maged1*-cKO thalamus samples (Fig. 3i). From these newly demonstrated

interactions we hypothesized that Maged1 plays a role in H2A mono-ubiquitination in thalamic neurons following cocaine intake, via Maged1 involvement with a PRC1 member. To test whether MAGED1 associates with USP7 in human cells, we performed co-IP-MS of MAGED1 in two different human cell lines: human embryonic kidney 293 T (HEK293T) and neuroblastoma (SH-SY5Y) cells (Fig. 3j–m and Supplementary Data 2). The interaction with USP7 was thus confirmed in human cells (Fig. 3l–m). In neuronal cells, the known interaction between MAGED1 and CREB[38] was also found (Fig. 3m). In addition, an interaction between MAGED1 and another H2A deubiquitinase, MYSM1, was also identified (Fig. 3m). These results indicated that the Maged1-USP7 axis is evolutionarily conserved and may regulate addiction behavior in humans through histone H2A post-translational modification (PTM).

## Cocaine induces an increase of histone H2A monoubiquitination in the NAc, PFC, and thalamus

We, therefore, tested whether chronic administration of cocaine impacts H2A monoubiquitination in key brain regions involved in drug addiction and whether these changes depend on Maged1 function (Fig. 4a). To this end, by WB, we analyzed H2A and H2Aub levels in the PFC, thalamus, and NAc of control and *Maged1*-(c)KO animals after chronic cocaine or saline administration. The results showed a significant increase in H2Aub in all these nuclei (Fig. 4b–e) after cocaine administration. Remarkably, the cocaine-dependent increase in H2Aub was only abolished in the thalamus of the *Maged1*-cKO mice (Fig. 4e). These results show that cocaine treatment induces an increase in H2Aub in key structures involved in addiction and that Maged1 is crucial for cocaine-induced H2A monoubiquitination, specifically in the thalamus.

## USP7 pharmacological inhibition or Maged1 inactivation precludes cocaine-evoked H2A monoubiquitination and cocaine sensitization

Because USP7 is a known PRC1 regulator with deubiquitinase activity[30–32], we hypothesized that USP7 plays a key role in the Maged1-dependent cocaine behavioral response. We therefore inhibited USP7 activity in vivo by administrating P5091, a selective USP7 inhibitor, in a two-injection cocaine sensitization protocol[14] (Fig. 4f). Interestingly, systemic injection of P5091 abolished locomotor sensitization in control mice, similar to the effect of *Maged1* inactivation (Fig. 4g, h and Supplementary Fig. 10), with no effect on spontaneous locomotor activity (Supplementary Fig. 10b), and no effect on the locomotor activity of *Maged1*-cKO mice (Supplementary Fig. 10c). To exclude a transient occlusion effect, we continued the experiment for 3 additional days of daily cocaine injections and observed a sustained abolition of cocaine sensitization (Supplementary Fig. 10d). Similarly, an intrathalamic injection of P5091 significantly interfered with cocaine-induced locomotor sensitization (Fig. 4i–j). Similar to the observation with *Maged1*-cKO mice, P5091 treatment prevented the increase in H2A monoubiquitination induced by cocaine administration (Fig. 4k). These results showed that pharmacological abolition of cocaine-evoked H2Aub deposition in the thalamus paralleled an alteration of cocaine-evoked behavioral sensitization.

## Cocaine-evoked histone H2A monoubiquitination and its subsequent transcriptional repression

To further characterize the mechanisms by which Maged1 activity and H2A mono-ubiquitination contribute to cocaine sensitization, we compared H2Aub genomic coverage in the thalamus of control and *Maged1*-cKO mice after chronic cocaine or saline were administered by performing chromatin immunoprecipitation with sequencing

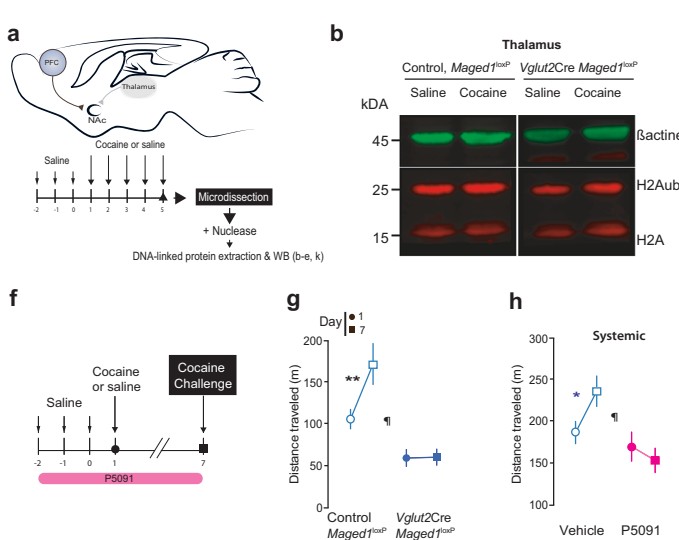

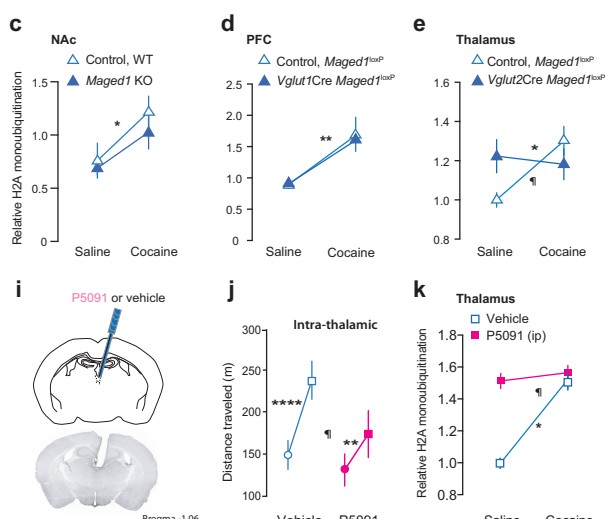

**Fig. 4 | Chronic cocaine administration increases H2A ubiquitination, which is impaired by *Maged1* inactivation or UPS7 inhibition. a** Method: cocaine administration (intraperitoneal (i.p.), 15 mg/kg) and nuclease-based protein extraction. **b** Representative western blot (thalamus). **c** H2A monoubiquitination (ratio between H2Aub and H2A signal intensities) in the NAc (**c**), PFC (**d**), and thalamus (**e**). In panel (**c**): control saline, n = 6; control cocaine, n = 5; cKO saline, n = 5; cKO cocaine, n = 4, (two-way ANOVA, control versus *Maged1* cKO, P = 0.3839; saline versus cocaine, *P = 0.0144; interaction factor, P = 0.6853). In panel (**d**): control saline, n = 9; control cocaine, n = 11; cKO saline, n = 10; cKO cocaine, n = 8, (two-way ANOVA, control versus *Maged1* cKO, P = 0.8751, saline versus cocaine, **P = 0.0023; interaction factor, P = 0.8428). In panel (**e**): control saline, n = 14; control cocaine, n = 20; cKO saline, n = 13; cKO cocaine, n = 18, (two-way ANOVA, control versus *Maged1*-cKO, P = 0.0849; saline versus cocaine, P = 0.4907; interaction factor, ¶P = 0.0246, Sidak's test (saline versus cocaine), control *P = 0.0281). **f** Cocaine sensitization scheme during USP7 inhibition (P5091, i.p., 10 mg/kg;

intrathalamic injection, 0.1 mg/kg and 400 nL). Cocaine-induced sensitization, control, n = 9; cKO, n = 8 (**g**), vehicle, n = 10; P5091, n = 8 (**h**), after paraventricular thalamus (PVT) P5091 injections, control, n = 7; P5091, n = 8 (**j**) (two-way ANOVA, control versus cKO, P = 0.0007; days, P = 0.0216; interaction factor (genotype x days) ¶P = 0.0249; Sidak's test (D1 versus D7) **P = 0.0044 (**g**); vehicle versus P5091, P = 0.0224; days, P = 0.1790; interaction factor (P5091 x days) ¶P = 0.0125; Sidak's test (D1 versus D7) *P = 0.012 (**h**); vehicle versus P5091, P = 0.1987; days, P < 0.0001; interaction factor (P5091 x days) ¶P = 0.0265; Sidak's test (D1 versus D7), ****P < 0.0001, **P = 0.011 (**j**)). **i**, Cannula implantation for P5091 injections into the PVT. **k** Thalamic H2Aub, control saline, n = 10; control cocaine, n = 4; P5091 saline, n = 6; P5091 cocaine, n = 9 (two-way ANOVA, vehicle versus P5091, P = 0.0007; saline versus cocaine, P = 0.0171; interaction factor ¶P = 0.0414; Sidak's test (saline versus cocaine) *P = 0.0249 for control group). **h, j, k**, All control mice received vehicle injections. The distance traveled was recorded for 30 min. Source data are provided as a Source Data file.

(ChIPmentation)[39] (Fig. 5a). Analysis of the H2Aub coverage in these mice revealed a higher number of regions with this PTM in the cocaine-treated mice compared to saline-treated mice (13493 and 8174, respectively; +164.95 % in the cocaine- vs. the saline-treated mice) (Fig. 5b). Supporting a regulatory function of this PTM, the same increase was observed when only gene promoter regions (defined as 5 kb upstream and 2 kb downstream of the gene transcription start site (TSS)) were analyzed, and the largest fraction of H2Aub+ elements was located less than 5 kb upstream from the nearest TSS. Cocaine shifted this distribution significantly upward in the control but not in the *Maged1*-cKO mice (Fig. 5c). Specifically, 1271 genes with H2Aub at their promoters were found in the control mice in the cocaine treatment condition, while only 143 loci showed H2Aub specific marks after saline administration (Fig. 5d). In contrast, the total number of H2Aub marks and promoters was comparable between the cocaine-injected (583 genes) and saline-injected (340 genes) *Maged1*-cKO animals (Fig. 5d). Furthermore, 84% (1068 genes) of genes displaying H2Aub enrichment upon cocaine administration in control mice did not show enrichment in *Maged1*-cKO animals, indicating that the loss of *Maged1* deeply impacted the H2Aub dynamics associated with the cocaine response (Fig. 5e, f). Cocaine-dependent H2Aub+ genes in control mice were associated with changes in biological processes like ion transport, synaptic signaling, axonal guidance, neuron projection, and cytoskeleton/microtubule organization (Supplementary Fig. 11 and Supplementary Data 1) which were not found in *Maged1*-cKO mice (Supplementary Data 1). These results support the idea that Maged1 exerts its specific effects on the behavioral response to cocaine primarily through drug-induced PRC1 gene repression, as cocaine administration to control mice triggered coherent transcriptional and H2Aub dynamics (Supplementary Fig. 6d, 7, 11). Indeed, we found a

significant overlap, between RNAseq and ChIP data, in the genes with transcriptional downregulation and those with promoters enriched in H2Aub marks after cocaine administration (Fig. 5g). Interestingly, we found that *c1qL3*, encoding a signaling protein involved in synaptic projection and human CUD[40,41], was downregulated while displaying an increase in H2Aub on its promoter upon chronic cocaine administration, as well as *Hopx* and *Scml4*, important epigenetic regulators linked to PRC1.

## *MAGED1-USP7* modifies addiction risk and behavioral response to cocaine in humans

To assess the potential contributions of *MAGED1* and *USP7* to CUD directly in humans, we conducted a within-case *MAGED1* and *USP7* gene analysis with 351 consecutively recruited outpatients with cocaine addiction and genetically verified Caucasian ancestry. After demonstrating the absence of structural variants in/near *MAGED1* or *USP7*, we focused on 175 nucleotide polymorphisms (SNPs). We investigated their potential associations with seven CUD-related phenotypes reflecting all stages of disease progression, correcting for multiple testing. The outpatient population comprised 77 % of men aged 38 ± 9 years old. All the outpatients reported lifetime cocaine use and 72% of these men suffered from severe CUD (including to crack-cocaine) (Supplementary Table 1). Comorbid substance use disorders (SUDs) were tobacco smoking (current for 89% of the sample), alcohol (62%), opioid (61%), cannabis (64%), and/or benzodiazepine (41%) use disorders (Supplementary Table 1). Half the sample had three-lifetime SUDs or more (tobacco excluded). Patients carrying at least one alternate allele were compared to those carrying only reference allele(s). In the case of multiple statistical associations with a given phenotype, we retained lead SNPs most likely associated with the

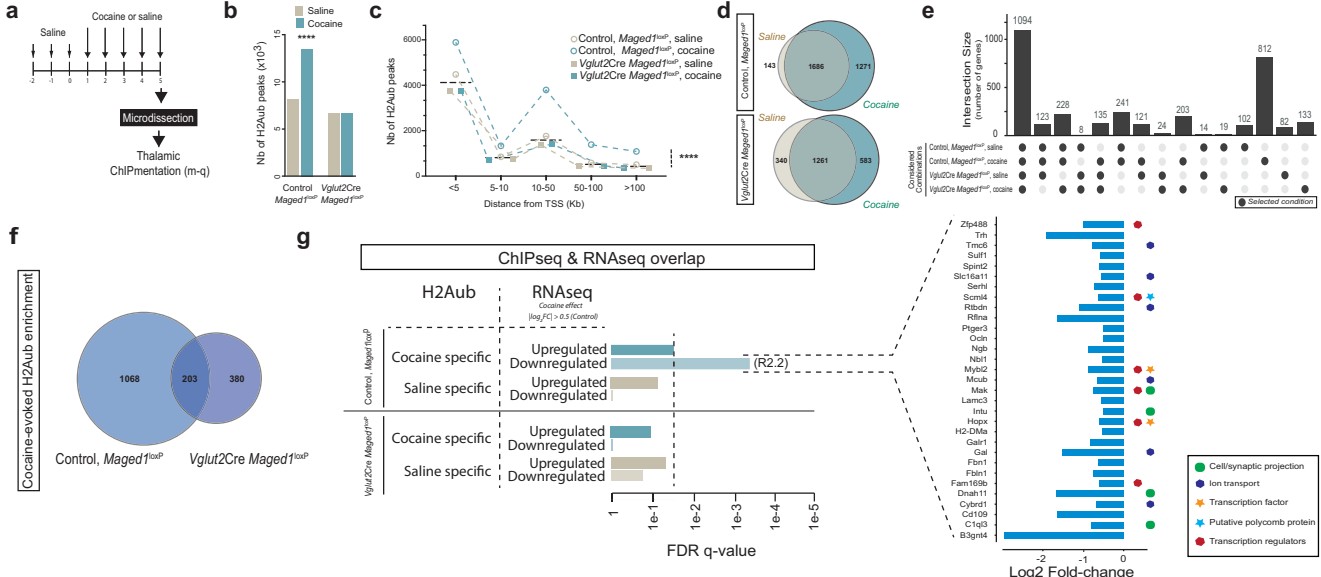

**Fig. 5 | Dynamic changes in thalamic H2A monoubiquitination and subsequent gene regulation in response to cocaine administration. a** Experimental timeline before thalamus microdissection for ChIPmentation. Plots representing the number of H2Aub+ (H2AK119ub) elements identified across different experimental conditions (**b**) and their distribution with respect to the nearest gene transcription start site (TSS) (**c**). The blue dashed line in panel (**c**) represents the median of 4 different experimental conditions. Repeated cocaine injection was associated with an increase in the total number of H2Aub+ elements compared to the saline condition; this increase was not observed in *Maged1*-cKO mice (chi-square, two-sided, ****P < 0.0001). Euler plots (**d**, **f**) and Upset (**e**) representing the number of H2Aub+ genes (displaying H2Aub+ elements at their promoters) across the different experimental conditions. Cocaine injection in control mice increased H2Aub+ genes (**d**, top). In contrast, in *Maged1*-cKO mice, the number of cocaine-specific

H2Aub + genes was considerably lower than that in control mice (**d**, bottom), and only a minor fraction of cocaine-specific H2Aub+ genes were shared between control and *Maged1*-cKO mice (**f**). The Upset plot (**e**) represents the number of H2Aub+ genes in common between the different experimental groups; bars represent the number of H2Aub+ genes in common (Intersection size) between the groups indicated below each of them with a dark gray circle. **g** Bar plot presenting the overlapping genes displaying H2Aub enrichment in saline or cocaine condition (in *Maged1* control and cKO mice) and up- or downregulated expression in *Maged1* control mice (log2|FC| ≥ 0.5, false discovery rate (FDR) < 0.05). A significant overlap was found between genes displaying H2Aub enrichment and those that are downregulated after cocaine administration. These genes are highlighted on a bar plot with their associated log2FC in control mice on the right. Source data are provided as a Source Data file.

underlying biological signal by combining positional and functional data. First, the transition from first cocaine use to CUD was significantly associated with 17 SNPs in *MAGED1* (Fig. 6a, b), of which we identified two lead SNPs. These two SNPs showed opposite effects during the transition, as highlighted by Kaplan−Meier survival curves (Fig. 6c). An in silico analysis with human cells showed a significant association of these two SNPs with DNA methylation changes and thus altered gene expression. Second, we observed associations between 44 *USP7* SNPs and Scale for Assessment of Positive Symptoms (SAPS)-aggression score (Fig. 6a), of which we identified five lead SNPs. Four SNPs were in the alternate allele associated with lower SAPS-aggression score, and scores were higher for the alternate allele of the other SNP (Fig. 6c). Consistent with our preclinical data, these findings with human samples revealed that *MAGED1* and *USP7* polymorphisms are associated with CUD (Supplementary Table 2). These associations were independent of potential confounders, such as biological sex and the self-reported amount of cocaine use, the latter of which is strongly related to lower SAPS-aggression score and childhood hyperactivity,

and is associated with a shorter transition from first cocaine use to CUD (Supplementary Results, Supplementary Tables 3 and 4). These data enabled us to designate *MAGED1* and *USP7* as independent contributors to the vulnerability for CUD and specific behavioral responses to cocaine.

## Discussion

Here we demonstrated that the major effect of Maged1 on drug-adaptive behavior is mediated outside of the mesocorticolimbic pathway through Vglut2 neurons in the PVT. Furthermore, we showed that the role played by Maged1 is bidirectional in the PVT of adult mice, with significant abolition and restoration of cocaine sensitization. Taking advantage of this phenotype, we found that *Maged1* inactivation significantly altered cocaine-evoked gene repression in the thalamus and further identified its involvement in PRC1 complex activity. The identification of USP7, a well-known PRC1 regulator, and H2Aub as Maged1-interacting partners, together with the observation that chronic cocaine exposure leads to an

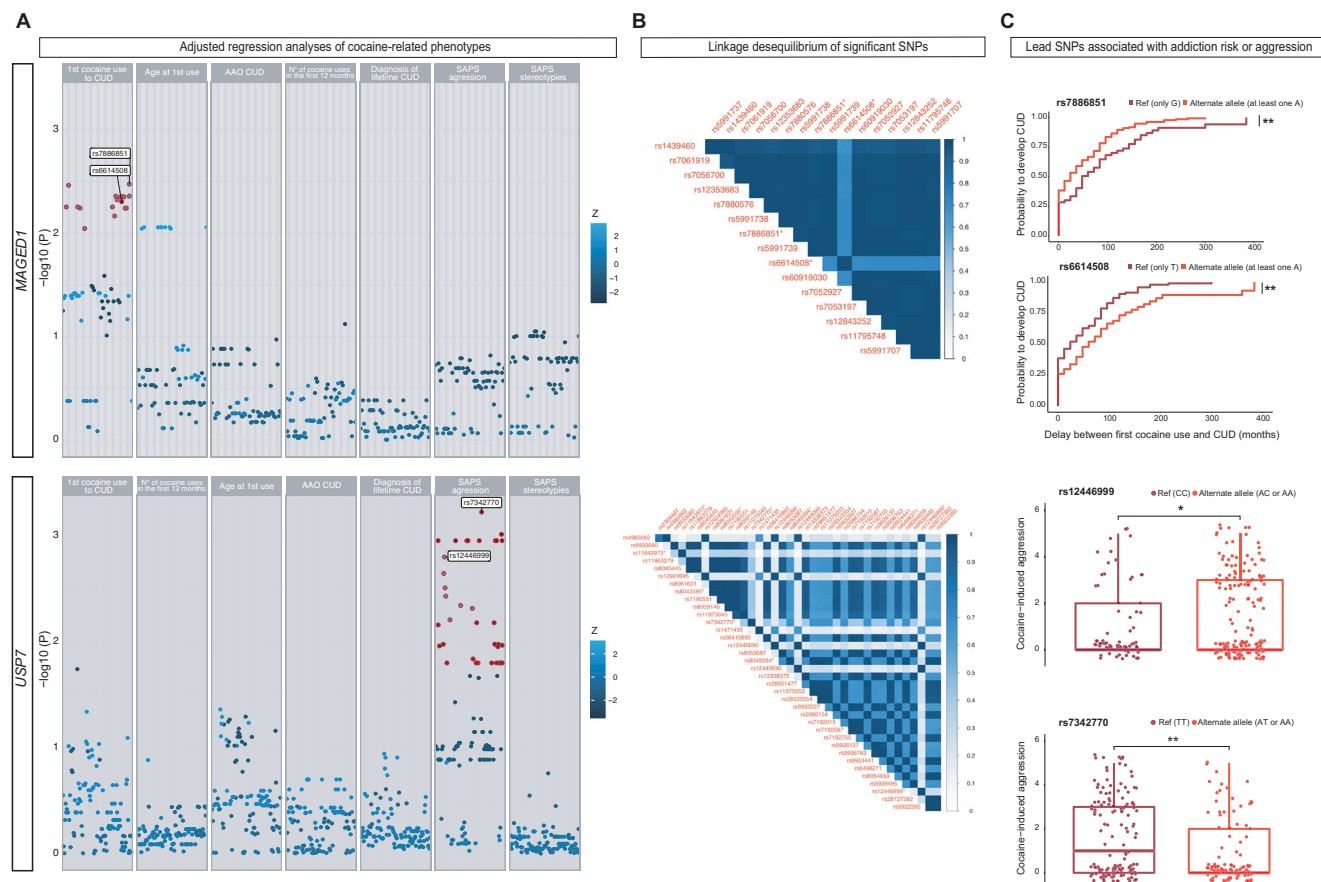

**Fig. 6 | Cocaine use disorder (CUD) is associated with small *MAGED1* and *USP7* polymorphisms in polydrug users. a** Scatterplots representing -log (p values) on the y-axis and effect sizes (Z) as the point color for 61 *MAGED1* single nucleotide polymorphisms (SNPs, upper panel) and 114 *USP7* SNPs (lower panel) tested using regression analyses with seven cocaine-related phenotypes (Cox regression for age at first cocaine use and transition from first cocaine use to cocaine user disorder (CUD), linear for the other; two-tailed tests), adjusted for sex and the first 10 ancestry components of up to 351 patients with strict Caucasian ancestry. Significant SNPs are circled in red, and the lead SNPs in each phenotype are labeled. *P* values were corrected using the Benjamini-Hochberg False Discovery Rate (within-phenotype). **b** Linkage disequilibrium heatmaps (European population from 1000 genomes reference, hg19 version) showing increasing r² values, as indicated from white to dark blue with 16 *MAGED1* SNPs (data were not available for one SNP) significantly associated with the transition from first cocaine use to CUD (upper

panel) and 37 *USP7* SNPs (data were not available for seven SNPs) significantly associated with lower SAPS-aggression score (lower panel). SNPs are ordered according to their genomic position. Lead SNPs are marked with an asterisk. **c** First upper panel - Kaplan−Meier survival curves for *MAGED1* rs7886851-A, which is associated with the transition from first cocaine use to CUD. Log-rank two-tailed test, **P = 0.002. Second upper panel - Kaplan−Meier survival curves for *MAGED1* rs6614508-A, which is associated with the transition from first cocaine use to CUD. Log-rank test, **P = 0.002. First lower panel − boxplot showing *USP7* rs12446999-A with lower SAPS-aggression score. N = 237. Wilcoxon−Mann−Whitney two-tailed test *P = 0.028. Second lower panel - boxplot showing *USP7* rs7342770 with lower SAPS-aggression score. N = 237. Wilcoxon-Mann-Whitney **P = 0.0014. Adenosine (A) is the alternate allele. AAO age at onset, CUD cocaine use disorder, SAPS Scale for Assessment of Positive Symptoms−Cocaine-Induced Psychosis. Source data are provided as a Source Data file.

overall increase in H2Aub+ promoters, indicated that Maged1 mediates cocaine-dependent gene downregulation by controlling histone H2A monoubiquitination. Furthermore, the inactivation of *Maged1* in Vglut2 neurons or inhibition of USP7 alters H2A mono-ubiquitination in the thalamus and cocaine-adaptive behavior. Considering that Maged1 enabled interactions between H2Aub and USP7, the Maged1-USP7 interaction likely constitutes a molecular switch linking PRC1-mediated gene repression to upstream signaling events[30].

This work also provides a genome-wide assessment of H2Aub deposition and transcriptional alterations in the thalamus in response to repeated cocaine administration. A striking finding of the present study is that, in this non-canonical region, the predominant effect of cocaine is gene repression, conversely to what has been observed in the NAc or the VTA[9].

Moreover, we observed a significant overlap between cocaine-induced repressed genes and cocaine-induced increase in H2Aub on the promoter, further demonstrating that this epigenetic mechanism has an overall significant impact on gene transcription level while controlling the predominant transcriptomic effect.

Even though evidence suggests that PRC1 complexes are targeted by PTMs, it is still unclear how H2Aub is regulated[42]. Here, we observed that among genes showing an increase in H2Aub mark on their promoter and significant repression induced by cocaine, at least 2 genes, *Hopx* and *Scml4*, are linked to the PRC1 complexes and may thus represent a regulatory feedback loop.

Our proposed model (Supplementary Fig. 12) postulates that when cocaine is administered, Maged1-USP7 is recruited to H2A, activating or stabilizing PRC1 and leading to H2A monoubiquitination. Thus, in *Maged1*-cKO mice or after treatment with the USP7 inhibitor P5091, PRC1 activity is blocked, leaving it in an unresponsive state such that upstream events (such as cocaine exposure) do not induce its activity (Supplementary Fig. 12). This epigenetic mechanism highlighted in mice might be a first step toward developing CUD in humans. Indeed, SNP mutations in *USP7* and *MAGED1* among polydrug users were associated with varying scores of aggressiveness during cocaine use and a modified transition time to cocaine addiction, respectively. Although no medication has been developed to be an efficacious treatment for CUD, a chronic disease with harmful consequences worldwide[43,44], the finding that a pharmacological inhibitor of USP7 is behaviorally effective for a cocaine-relevant behavior in mice might pave the way for the development of new treatments for CUD. The lack of characterization of small-molecule inhibitors and their targeted deubiquitinases is a limiting factor in the development of efficient and selective deubiquitinase inhibitors that could be used in human patients[45]. However, this is now likely to change with new reports of selective USP7 inhibitors that possess a structurally defined mechanism of inhibition[46,47].

In conclusion, these findings establish a critical role for histone H2A monoubiquitination and its subsequent transcriptional changes, in regulating cocaine-adaptive behaviors, which may constitute an essential epigenetic risk factor for drug addiction in humans. Taken together, our ChIP, RNAseq, and human genomic analyses position H2Aub in the drug-related epigenetic landscape and link this histone mark to an acquired human disease. The regulation of this newly identified cocaine-induced epigenetic modification might be considered a potential therapeutic target for prevention and harm reduction in active cocaine users.

## Methods

Experimental procedures were performed in accordance with the Animal Care Committee guidelines and approved by the ethical committee of the Faculty of Medicine of the Free University of Brussels.

### Animals

The *Maged1*-inactivated (*Maged1*-KO) and conditioned (*Maged1*<sup>loxP</sup>) mice used in the present study have been previously generated and described[48]. *Maged1*-KO do not show gross abnormalities and have normal general cognitive and sensorimotor abilities[10,48–50]. All our experiments were performed in 2- to 4-month-old mice. Our experiments were conducted on hemizygous *Maged1*-KO males and their wild-type littermates (*Maged1* WT) generated by crossing heterozygous *Maged1* KO/WT females with C57Bl6J males as *Maged1*-KO males are deficient in sexual behavior and do not reproduce properly[50]. Conditional knockout mice were generated by crossing heterozygous *Maged1*<sup>loxP/WT</sup> or homozygous *Maged1*<sup>loxP/loxP</sup> females with different population selective Cre male mice. The Cre-mediated deletion of *Maged1* was obtained using mice expressing Cre-recombinase under the control of the *Vglut2* promoter, *Vglut1* promoter, and *Emx1*[51] promoter. As with the *Maged1*-KO mice, we used hemizygous *Maged1*-cKO males and their floxed littermates to maintain the same conditions allowing comparison of the KO and cKO types. The *Vglut1*-IRES2-Cre (Stock No: 023527) and *Vglut2*-ires-Cre (Stock No: 016963) were obtained from the Jackson laboratory. All mice were backcrossed for at least eight generations in the C57Bl6J genetic background. Mice were housed in groups of 2–4 mice, in a 12 h light/dark cycle, with food and water available ad libitum.

### Genotyping

We systematically verified the genotype of each animal by PCR on DNA extracted from ear biopsies (1–2 mm). A list of the primers used for the wild-type *Maged1* and *Maged1*-KO and *loxP* alleles are found in de Backer et al. [10] For the *Vglut1*Cre allele, the *Vglut1* mutant (CCC-TAG-GAA-TGC-TCG-TCA-AG) and common (ATG-AGC-GAG-GAG-AAG-TGT-GG) primers give an amplicon of 344 bp. For the *Vglut2*Cre allele, the *Vglut2* mutant (ATC-GAC-CGG-TAA-TGC-AGG-CAA) and common (CGG-TAC-CAC-CAA-ATC-TTA-CGG) primers give an amplicon of 600 bp. For the *Emx1*Cre allele, the *Emx1* mutant (TCG-ATA-AGC-CAG-GGG-TTC) and common (CAA-CGG-GGA-GGA-CAT-TGA) primers give an amplicon of 195 bp.

### AAV stereotaxic injections

To inactivate *Maged1* in a region-specific manner we used *Maged1*<sup>loxP</sup> mice in which we stereotaxically injected an AAV containing the Cre-recombinase in the ventral subiculum and the PVT (from bregma, in mm): ventral subiculum, A-P -3,64, M-L + -3,15, D-V -4,72 (400 nL), PVT, A-P -1.46, M-L + -1.02, D-V -3.05, (500 nL) angle: 10°.

Behavior experiments after stereotaxic injection aiming at *Maged1* inactivation were controlled by AAV-eGFP injection in littermate mice. In specified cases, control mice were *Maged1* WT mice injected with AAV-eGFP-Cre.

Mice were anesthetized with avertin (2,2,2-tribromoethanol 1.25%; 2-methyl−2-butanol 0.78%, 20 ll/g, ip, Sigma-Aldrich) before bilateral injection of AAV on a stereotaxic frame (David Kopf Instruments). Injections were performed using a blunt needle connected to a 10 μL syringe (Hamilton) and a microstereotaxic injection system (KD Scientific). The injection rate was fixed at 0.1 μL/min. The needle was left in place for 15 min after the injection before being slowly removed. To avoid bias, littermates were equally distributed between the two groups. After surgery, mice were allowed to recover for at least 3 weeks.

We used serotype 5 known for its neuronal tropism, and lowest ability to migrate anterogradely[52].

Vectors used in this study were purchased from Penn Vector Core (University of Pennsylvania):

AAV5.CMV.HI.eGFP-Cre.WPRE.SV40 (qPCR-based genome copy titer: 1.864e12)

AAV5.CMV.PI.eGFP.WPRE.bGH (qPCR-based genome copy titer: 1.704e12)

To re-express *Maged1* in a region-specific manner we used *Maged1*-KO mice in which we stereotaxically injected an AAV containing *Maged1* and an *eGFP* (linked with a P2A sequence), either in the PFC, in the amygdala (PFC, A-P 1.9, M-L + -0.3, D-V −2.3 (600 nL); amygdala, A-P -1.0, M-L + -3.0, D-V -4.9 (600 nL)); or in the PVT. The vectors were generated by transient transfection (with X-tremeGENE) of HEK293 cells using three plasmids: our custom-made plasmid, cis ITR-containing CMV-eGFP-P2A-Maged1 (or just CMV-eGFP), the trans-plasmid encoding AAV replicase and capsid genes (pAAV.DJ/8) and the adenoviral helper plasmid followed by purification (with freezing and thawing and centrifugation) and qPCR-based genome copy titer quantification.

## Cocaine-induced locomotor sensitization

Experiments were performed during the light period of the cycle. To evaluate spontaneous locomotor activity, mice were allowed to freely explore a wide open-field (OF; $40 \times 40 \times 40$ cm), made of gray (wall) and white (flour) hard plastic, for 30 to 60 min (as specified). Locomotor activity was recorded in a low luminosity environment (18–22 lx) with a video camera placed on the ceiling above the arena and connected to an automated video tracking system (Ethovision XT 13-14, Noldus Information Technology). Classical cocaine-induced locomotor sensitization: the three-first days of the experiment, mice were injected with saline and their locomotor activity was recorded, the next 5 days, mice were injected with cocaine instead of saline (20 mg/kg, intraperitoneal (i.p.)). When specified, we performed a two-injection protocol (15 mg/kg, i.p.) with 3 days of habituation and 7 days between the two cocaine injections, based on Valjent et al. work[14].

## Cocaine-induced CPP and operant self-administration

Conditioned place preference was conducted in a three chambers apparatus, consisting of a small middle neutral ($6 \times 20$ cm) area connected to two large compartments ($18 \times 20$ cm) that differed in wall and floor conditions (Panlab). Three days before the experiment, mice were used to be manipulated and received a single saline injection in their home cage. At day 0 (preconditioning test, 18 min), mice were placed in the central neutral area and allowed to explore freely both chambers. Mice were randomly assigned to the various experimental groups (unbiased protocol). Conditioning (days 1–6) was carried out as follows: mice were confined to one compartment for 30 min immediately after injection of cocaine (10 mg/kg, ip, or saline for control groups) on days 1, 3, and 5 and to the other compartment after saline injection on days 2, 4 and 6. For the post-conditioning test (day 7), mice were placed in the central area and allowed to explore freely both chambers for 18 min.

For operant self-administration, mice were first anesthetized by a mixture of ketamine hydrochloride (Imalgène; Merial Laboratorios S.A., Barcelona, Spain; 75 mg/kg) and medetomidine hydrochloride (Domtor; Esteve, Barcelona, Spain; 1 mg/kg) dissolved in sterile 0.9% physiological saline. This anesthetic mixture was administered intraperitoneally in an injection volume of 10 mL/kg of body weight. After surgery, a subcutaneous injection of atipamezole hydrochloride (Revertor; Virbac, Barcelona, Spain; 2.5 mg/kg of body weight) was administered to reverse the anesthesia. Additionally, mice received a subcutaneous injection of meloxicam (Metacam; Boehringer Ingelheim, Rhein, Germany; 2 mg/kg of body weight) and an intraperitoneal injection of gentamicin (Genta-Gobens; Laboratorios Normon, S.A., Madrid, Spain; 1 mg/kg of body weight) all previously dissolved in sterile 0.9% physiological saline. For operant conditioning maintained by cocaine, cocaine hydrochloride (Sigma-Aldrich, Saint Louis, MO, USA) was dissolved in a saline solution (0.9% NaCl w/v). The catheter's patency was evaluated by thiopental sodium (5 mg/mL) (Braun Medical S.A, Barcelona, Spain) dissolved in distilled water and delivered by infusion of 0.1 mL through the intravenous catheter. The experimental sequence was the following: mice were trained under a fixed ratio 1 schedule of reinforcement (FR1; one nose-poke produces the delivery of one dose of cocaine) during 5 consecutive sessions (sessions 1–5) followed by 5 sessions (sessions 6-10) under an FR3 schedule (three nose-pokes produce the delivery of one dose of cocaine).

## Immunohistochemistry

Mice were transcardially perfused with a 0.01 M PBS solution followed by paraformaldehyde 4% in 0.1 M phosphate buffer (pH 7.4). Brains were removed, fixed overnight in paraformaldehyde 4%, and successively cryoprotected in 20 and 30% sucrose in 0.1 M phosphate buffer. 20-um-thick coronal sections were cut on a cryostat (−22 °C). Sections were incubated in 10% normal horse serum in 0.01 M PBS-0.1% Triton X-100 for 60 min at room temperature to block the nonspecific antibody binding. Sections were then incubated overnight at 4 °C with the following required primary antibodies: chicken GFP (1:2000, Abcam, ab13970), rabbit mono-ubiquitylated histone H2A (1:1500, Cell Signaling, 8240 S), goat USP7 (1/2000, Bethyl Laboratories, A303-943A), mouse mono-ubiquitylated histone H2A (1:75, Merck, 32160702), Maged1 (1/300, Cosmo-Bio, BAM-74-112-EX), in 1% normal horse serum, 0.01 M PBS and 0.1% Triton X-100.

After three washes with PBS/0.1% Triton X-100, slices were incubated in PBS for 1 h at room temperature and incubated for 2 h at room temperature with the appropriated Alexa488 (1/400, Invitrogen), Alexa594 (1/400, Invitrogen) Alexa647 (1/400, Invitrogen) secondary antibodies. Hoechst 33342 (1/5000, H3570, Invitrogen) was used (5 min and washed twice in PBS) for nuclear staining. The sections were next mounted on a Superfrost slide (Thermo Scientific) and dried using a brush before adding Glycergel mounting medium (Dako). Imaging was performed using a Zeiss LSM780 confocal microscope, an AxioImager Z1 (Zeiss), and a wide-field Fluorescent Microscope: Zeiss AxioZoom.v16 all controlled by the Zen Black software (Zeiss).

## In situ hybridization autoradiography

Mice were deeply anesthetized using halothane and killed by dislocation and decapitation, and the whole brain was rapidly removed and frozen on dry ice. Coronal sections (18 μm thick) were prepared with a cryostat (CM 3050, Leica), mounted onto RNase-free glass slides (SuperfrostPlus, Thermo Scientific), and stored at -20 °C before the experiment. In situ hybridization experiments were performed as previously described[10,53].

## Fluorescent In situ hybridization with RNAscope

RNAscope ISH was used to detect cell type-specific expression of *Vglut1* mRNA, *Vglut2* mRNA, and *Maged1* mRNA. Mice were deeply anesthetized, and the whole brain was removed and rapidly frozen on dry ice. Fresh-frozen tissue sections (16 μm thick) were mounted on positively charged microscopic glass slides (Fisher Scientific) and stored at −80 °C until RNAscope ISH assays were performed. Multiple target gene-specific RNAscope probes were used to observe the cellular distributions of *Vglut1*, *Vglut2* and *Maged1* in the whole brain by using a *Vglut2* RNAscope probe (Mm-Slc17a6-C2-Vglut-2 probe, cat. no. 319171-C2, targeting 1986–2998 bp of the Mus musculus *Vglut2* mRNA sequence, NM_080853.3), a custom-made *Maged1* RNAscope probe (Mm-Maged1-O1, cat. no. 563041, targeting 901-1792 bp (targeting exon 4-12) of the Mus musculus *Maged1* mRNA sequence, NM_019791.3), *Vglut1* RNAscope probe (Mm-Slc17a7-C3-Vglut1 probe, cat. no. 416631-C3, targeting 464 - 1415 bp of the Mus musculus *Vglut1* mRNA sequence, NM_182993.2). TSA fluorophores, cyanine3 (NEL744001KT), cyanine5 (NEL745001KT), and fluorescein (NEL741001KT), were obtained separately at PerkinElmer. All these probes were designed and provided by Advanced Cell Diagnostics. The RNAscope mRNA assays were performed following the manufacturer's protocols. Stained slides were coverslipped with fluorescent mounting medium (ProLong Gold anti-fade reagent P36930; Life Technologies) and imaged with an AxioImager Z1 (Zeiss) microscope at 20-40x

magnification with the use of Zen Black software (Zeiss). Cells expressing *Vglut1* mRNA, *Vglut2* mRNA, and *Maged1* mRNA in the cortex (PFC, motor (M1), and somatosensorial cortex) and the PVT were counted in eight stereologically selected sections per brain at 40x magnification.

### Flow cytometry and cell sorting

Animals were deeply anesthetized using halothane and killed by dislocation and decapitation. Brains were rapidly removed and immersed in ice-cold PBS where the whole thalamus was dissected[54,55]. We then performed a manual dissociation in an ice-cold dissociation medium (98 mM $Na_2SO_4$, 30 mM $K_2SO_4$, 5.8 mM $MgCl_2$, 0.25nmM $CaCl_2$, 1 mM HEPES pH 7.4, 20 mM glucose, 125 μM NaOH)[56] and filtered the cells through a 40 μm nylon mesh right before sorting. Cell sorting was performed using a FACS Aria III cell sorter supported by the FACSDiva software (BD Biosciences). Cells were selected by forward scatter, side scatter, Hoechst dye exclusion, and red fluorescence. Cell sorting was carried out at the purity mode of sort precision using a low sheath pressure with a 70 μm nozzle and sorted cells were directly collected into ice-cold phenol and guanidine thiocyanate (QIAzol, QIAGEN) for subsequent RNA extraction. For the assessment of sort quality, cells were sorted in ice-cold PBS and re-analyzed. Total RNA extraction and DNase treatment were carried out using miRNeasy Micro Kit and RNase-free DNase set according to the manufacturer's instructions (QIAGEN).

### Transcriptome analyses

Total mRNA from six sorted independent samples (3 *Vglut2*Cre, loxp-stop-loxp-tdTomato, *Maged1*<sup>loxP</sup> and 3 *Vglut2*Cre, loxp-stop-loxp-tdTomato, *Maged1* WT) and 15 unsorted independent samples (*Vglut2*Cre::*Maged1*<sup>loxP</sup> or WT, 8 with chronic cocaine, 7 with saline injections) was extracted using the QIAGEN miRNeasy micro kit according to the manufacturer's recommendations. A highVglut2 neurons proportion invited us to skip the FACS procedure for further transcriptomic experiment (Fig. 3) to avoid any transcriptomic modification relative to the sorting process, and we immediately mechanically dissociated the cells in phenol and guanidine thiocyanate (QIAzol, QIAGEN) which facilitate lysis of fatty tissues and inhibit RNases. Following RNA quality control assessed on a Bioanalyzer 2100 (Agilent technologies), the indexed cDNA libraries were prepared using the NEBNext rRNA Depletion Kit (New England BioLabs), followed by NEBNext Ultra II Directional RNA Library Prep, following the manufacturer's recommendations. The multiplexed libraries (18 pM) were loaded on flow cells and sequenced with a 20 million reads depth, 2×100 bases paired-end (Novaseq 6000, Illumina).

Approximately 20 million paired-end reads per sample were mapped against the mouse reference genome (GRCm38.p4/mm10) using STAR 2.5.3a software[57] to generate read alignments for each sample. Annotations Mus_musculus.GRCm38.87.gtf were obtained from ftp.Ensembl.org. Read counts and transcript per million reads (TPMs) were generated using tximport R package version 1.10.0 and lengthScaledTPM method (Soneson et al.[58]) with inputs of transcript quantifications from tool salmon (Patro et al.[59]). Low-expressed transcripts and genes were filtered based on analyzing the data mean-variance trend. The expected decreasing trend between data mean and variance was observed when expressed transcripts were determined as had ≥ 3 of the 40 samples with count per million reads (CPM) ≥ 1, which provided an optimal filter of low expression. A gene was expressed if any of its transcripts with the above criteria were expressed. The TMM method was used to normalize the gene and transcript read counts to $log_2$-CPM (Bullard et al.[60]). The principal component analysis (PCA) plot showed the RNA-seq data did not have distinct batch effects. Limma R package was used for 3D expression comparison (Ritchie et al.[61]; Law et al.[62]). To compare the expression changes between conditions of experimental design, the contrast

groups were set as Ctrl.Saline-Ctrl.Cocaine, KO.Saline-KO.Cocaine, Ctrl.Saline-KO.Saline, Ctrl.Cocaine-KO.Cocaine, (Ctrl.Cocaine-Ctrl.Saline)-(KO.Cocaine-KO.Saline). For DE genes/transcripts, the $log_2$ fold change ($L_2FC$) of gene/transcript abundance were calculated based on contrast groups, and the significance of expression changes was determined using a t test. P values of multiple testing were adjusted with BH to correct the false discovery rate (FDR) (Benjamini and Yekutieli[63]). A gene was significantly DE in a contrast group if it had adjusted p value < 0.05.

### Protein extraction

On the fifth day of cocaine (or saline) injection, 5 min after injection, mice were deeply anesthetized using halothane and killed by dislocation and decapitation. Brains were rapidly removed and immersed in ice-cold PBS where PFC, NAc, and whole-thalamus were dissected. The tissue was mechanically dissociated using an electric rotating pestle (Ultra-Turrax, Merck) and cells were lysed in lysis buffer (before IP-MS and parallel WB: 50 mM tris HCL pH 8.0, 150 mM NaCl, 1 mM EDTA, 0.5 % NP-40; for subsequent IP-WB focusing on histones: 10 mM Tris-HCl pH 8.0, 300 mM sucrose, 300 mM NaCl, 0.5 % NP40) for 25 min on ice with protease inhibitor (Complete, Roche, diluted as recommended by the manufacturer) and phosphatase inhibitor (PhosSTOP, Roche, diluted as recommended by the manufacturer). For IP-MS, after centrifugation at 4 °C for 20 min, at 15,000 $g$, only the supernatant was used. For histone-related subsequent IP-WB, centrifugation at 4 °C at 1200 × $g$ for 5 min was done to separate both cytosolic and nuclear soluble proteins (in the supernatant) from chromatin-linked proteins (in the pellet). The pellet was treated with a nuclease (Benzonase, Merck, cat. no. E1014) to release mononucleosomes[64] and related protein interactions. Proteins were quantified using a Bradford assay (ThermoFisher, cat. no. 23246).

### Immunoprecipitation with mouse lysates

Dynabeads Protein A (30 μL, Invitrogen, cat. no. 10001D) were washed four times in protein extraction lysis buffer (without protease or phosphatase inhibitors). The beads were then incubated with the IP antibody (2 μL for Maged1, 7 μL for H2Aub, and 7 μL for USP7 were used per sample) for 2 h at room temperature. The beads were subsequently washed 4 times before proteins were added (500 μg of proteins was added for IP-MS, 1.5 mg was used for IP-WB). Samples were incubated overnight with rotation at 4 °C.

For MS, beads were washed by the lysis buffer one time, and three times in trypsin digestion buffer (20 mM Tris-HCl pH 8.0, 2 mM CaCl2) and resuspended in 150 μL trypsin digestion buffer.

For WB, beads were washed in the lysis buffer four times before being resuspended in 40 μL of 2X SDS buffer and thawed for 15 min at 75 °C. The beads were then removed using the magnetic rack.

### Mass spectrometry from mouse lysates

Liquid chromatography–tandem mass spectrometry was done in Francis Impens Lab (VIB Center for Medical Biotechnology, UGent). After pull down, the washed beads were resuspended in 150 μL trypsin digestion buffer and incubated for 4 h with 1 μg trypsin (Promega) at 37 °C. Beads were removed, another 1 μg of trypsin was added and proteins were further digested overnight at 37 °C. Peptides were purified on SampliQ SPE C18 cartridges (Agilent), dried and re-dissolved in 20 μL loading solvent A (0.1% trifluoroacetic acid in water/acetonitrile (ACN) (98:2, v/v)) of which 2 μL was injected for LC-MS/MS analysis on an Ultimate 3000 RSLC nano LC (Thermo Fisher Scientific, Bremen, Germany) in-line connected to a Q Exactive mass spectrometer (Thermo Fisher Scientific). Peptides were first loaded on a trapping column made in-house, 100 μm internal diameter (I.D.) × 20 mm, 5 μm beads C18 Reprosil-HD, Dr. Maisch, (Ammerbuch-Entringen, Germany) and after flushing from the trapping column the peptides were separated on a 50 cm μPAC™ column with C18-endcapped functionality

(Pharmafluidics, Belgium) kept at a constant temperature of 35 °C. Peptides were eluted by a stepped gradient from 98 % solvent A' (0.1 % formic acid in water) to 30% solvent B' (0.1% formic acid in water/acetonitrile, 20/80 (v/v)) in 75 min up to 50 % solvent B' in 25 min at a flow rate of 300 nL/min, followed by a 5 min wash reaching 99% solvent B'. The mass spectrometer was operated in data-dependent, positive ionization mode, automatically switching between MS and MS/MS acquisition for the five most abundant peaks in a given MS spectrum. The source voltage was 3.0 kV, and the capillary temperature was 275 °C. One MS1 scan (m/z 400 – 2,000, AGC target 3 ×E6 ions, maximum ion injection time 80 ms), acquired at a resolution of 70,000 (at 200 m/z), was followed by up to 5 tandem MS scans (resolution 17,500 at 200 m/z) of the most intense ions fulfilling predefined selection criteria (AGC target 50.000 ions, maximum ion injection time 80 ms, isolation window 2 Da, fixed first mass 140 m/z, spectrum data type: centroid, intensity threshold 1.3xE4, exclusion of unassigned, 1, 5-8, >8 positively charged precursors, peptide match preferred, exclude isotopes on, dynamic exclusion time 12 s). The HCD collision energy was set to 25% Normalized Collision Energy and the polydimethylcyclosiloxane background ion at 445.120025 Da was used for internal calibration (lock mass). QCloud was used to control instrument longitudinal performance during the project[65].

Analysis of the mass spectrometry data was performed in MaxQuant (version 1.6.11.0) with mainly default search settings including an FDR set at 1% on PSM, peptide, and protein level. Spectra were searched against the Mouse proteins in the UniProt/Swiss-Prot reference database (release version of June 2019 containing 22,282 mouse protein sequences, UP000000589, downloaded from http://www.uniprot.org), supplemented with the protein A recombinant staphylococcus aureus protein sequence. The mass tolerance for precursor and fragment ions was set to 4.5 and 20 ppm, respectively, during the main search. Enzyme specificity was set as C-terminal to arginine and lysine, also allowing cleavage at proline bonds with a maximum of two missed cleavages. Variable modifications were set to oxidation of methionine residues, and acetylation of protein N-termini. Matching between runs was enabled with a matching time window of 0.7 min and an alignment time window of 20 min. Only proteins with at least one unique or razor peptide were retained. Proteins were quantified by the MaxLFQ algorithm integrated into the MaxQuant software. A minimum ratio count of two unique razor peptides was required for protein quantification. Further data analysis was performed with the Perseus software (version 1.6.2.1) after loading the protein groups file from MaxQuant.

## Western blot

Samples (input (eluted in 2X SDS buffer and thawed 15 min at 75 °C) and IP proteins) were run in NuPAGE 4%–12% Bis-Tris Mini Protein Gel (ThermoFisher Scientific, NP0321BOX) at the voltage of 100 V for 2.5 h in a NuPAGE MOPS SDS Running buffer (ThermoFisher, cat. no. NP0001) and then transferred to Nitrocellulose Blotting Membrane (Bio-Rad, cat. no. 1704270, 0.2 μm) and placed in a Trans-Blot Turbo Transfer System (Bio-Rad) (1.3 A constant and up to 25 V) for 10 min. PageRuler Plus Prestained Protein Ladder, 10 to 250 kDa was used (ThermoFisher, cat. no: 26619). The membrane was blocked in the intercept TBS blocking buffer (Li-Cor, cat. no. 927-60001) buffer for 1 h at room temperature and subsequently incubated in the blocking buffer containing rabbit Maged1, (1/3000, Cosmo-Bio, BAM-74-112-EX), goat USP7 (1/700, Bethyl Laboratories, A303-943A), rabbit H2A (1/1000, Merck, 07-146), and rabbit mono-ubiquitylated H2A (1:1000, Cell Signaling, 8240 S) antibodies overnight at 4 C.

For qualitative analysis, this was followed by the incubation in the blocking solution containing secondary antibody anti-rabbit or/and anti-goat IgG antibody conjugated with HRP (GENA934, AP106P, Merck) at room temperature for 1 h for enhanced chemiluminescence.

Pierce ECL Western Blotting Substrate (Cat. #32106, ThermoFisher Scientific) was used for signal detection.

For quantitative purposes, this was followed by the incubation in the blocking solution containing secondary antibody goat anti-rabbit DyLight 680 (ThermoFisher, cat. no. 35568) or/and donkey anti-goat DyLight 680 (ThermoFisher, cat. no. SA5-10090). For quantification, ImageStudioLite version 5.2.5 (LI-COR, 2014) was used. We defined regions of interest (ROIs) for each band of interest and performed background subtraction to remove any non-specific signals. We then measured the intensity of each band within the ROIs. We normalized the band intensities to the appropriate reference protein. We then imported the quantification data into a GraphPad Prism and performed two-way ANOVA to compare the different conditions.

The raw representative WB gel and immunoblot following IP are shown in the Source Data File.

## Human cell lines

Human embryonic kidney 293 T (HEK293T) and human neuroblastoma cells (SH-SY5Y) were cultured in Dulbecco modified Eagle's medium containing 10% fetal bovine serum (FBS Qualified, Gibco) at 37 °C in an atmosphere of 5% CO2.

The *Maged1* KO in HEK cells was generated by using the RNA-guided CRISPR-Cas nuclease system[66]. For the design, the Synthego and E-CRISP bioinformatic tools were used to minimize off-target effects. The input target genomic DNA sequence used was the human genome GRCh38.p13 from ENSEMBL where the *Maged1* gene locates on ChrX p11.22. As a results of the analysis six 20-nucleotide guide sequence were selected between exon 2 and exon 5 (gRNA1=GGCAGGGGCGTTATACTACAG, gRNA2=CAAGGCGCTGTCTTCTACG, gRNA3=GCCTCAGCCTGCAAACACAG, gRNA4=AATGTTGAAGAGAACAGCAG, gRNA5=GTGGAATTGGCCAATCAGCG, gRNA6=GATGCTTAAGGACTACACAA). The 6 gRNAs were synthesized inside a U6RNA cassette between the U6 promoter and the gRNA scaffold by Twist Bioscience. Starting from the 6 cassettes nine different combinations of tandem gRNA were coupled to cut at the extremities of exon 2 and exon 5 (combo 1 = gRNA1 + 4, combo 2 = gRNA1 + 5, combo 3 = gRNA1 + 6, combo 4 = gRNA2 + 4, combo 5 = gRNA2 + 5, combo 6 = gRNA2 + 6, combo7 = gRNA3 + 4, combo 8 = gRNA3 + 5, combo 9 = gRNA3 + 6) and directly transfected in HEK 293 T cells with an empty Cas9 plasmid. HEK 293T cells were maintained according to the manufacturer's recommendations. Cells were cultured in DMEM medium supplemented with 10% (v/v) FBS at 37 °C and 5% CO2. The cells were plated onto 6-well plates without antibiotics 16-24 h before transfection and seeded at a density of 0.5 ×10⁶ per well in a total volume of 2 mL. Cells were transfected with polyethyleneimine (PEI) and Cas9 plasmid with the 2 gRNA mixed at equimolar ratios using 1 μg of total DNA (0.7 μg of pX459-Cas9-Puro + 96 ng for each gRNA), 2 μg/mL of puromycin was applied 24 h after transfection for 48 h. The clonal-density dilution procedure was then applied to isolate the clonal cell line and expanded for 2–3 weeks. Finally, the DNA was extracted by using Monarch® Genomic DNA Purification Kit (NEB, T3010S). For the analysis of microdeletions of the targeted exons an Out-Fwd (p1 GCACAGTCAGACCACAGTCACC) and Out-Rev primers (p2, GAGCCAACCATGGAAAGGAGGG) both of which were designed to anneal outside of the deleted region, were used to verify the successful deletion by product size. A parallel set of PCRs with In-Fwd (p1 same as above) and In-Rev primers (p3 CTGCGGTGGCCTGGTTAGTAG) was designed to screen for the presence of the WT allele. By analyzing the PCR result the combo 6 (gRNA 2 + 6) was chosen as the best combination to delete the region between exon 2 and 5, even if a wild-type band is still present. This was expected since HEK293T cells have 3 X chromosomes containing the *MAGED1* gene, so the heterozygous clone selected (clone A1) was re-transfected with Cas9 plasmid + gRNA 2 + 6, two isolated sub-colonies were obtained and the abolished expression of

MAGED1 at protein level was confirmed by WB (Fig. 3k). The sub-clone C1 was selected and used for subsequent experiments.

## Immunoprecipitation from human cells

For MAGED1 co-IP in HEK cells, a KO cell line for *MAGED1* was used as control, while in SH-SY5Y cells rabbit IgG-IP was used as isotype control with respect to MAGED1 antibody (polyclonal IgG rabbit). Wild-type HEK293T, *MAGED1* KO HEK293T, and SH-SY5Y cells were grown in triplicates in 150 mm dishes (20 ×10$^6$ cells). Cells were lysed with lysis buffer (50 mM Tris–HCl pH 7.5, 150 mM NaCl, 1% Triton X-100) and supplemented with protease inhibitors (Roche, cat. no. 11836170001). The cell lysate was incubated for 30 min at 4 °C, sonicated for 5 min at 0 °C, and then centrifuged at 15,000 × $g$ for 15 min. The total protein concentration was determined by BCA assay and 500 μg of total proteins was incubated for 1 h (4 °C, rotary agitation) with 25 μL of Pierce™ Protein A Agarose beads (ThermoFisher, cat. no. 20333), centrifuged at 2,500 $g$ for 3 min at 4 °C and the supernatant recovered as pre-cleared lysate. For the immune complex formation (total proteins + antibody) 500 μg of the pre-cleared lysate was incubated overnight (4 °C, rotary agitation) with 2.5 μg of MAGED1 antibody (ThermoFisher, cat. no. PA5-99091) or 2.5 μg of IgG rabbit (Biotechne, cat.no. NB810-56910) as isotype control. To reduce non-specific binding and background, 25 μL of protein A Agarose beads were incubated for 1 h with 0.1% of BSA (4 °C, rotary agitation) and washed two times with 500 μL of IP buffer (50 mM Tris–HCl pH 7.5, 150 mM NaCl). The pre-blocked protein A beads were added to the immune complex at 4 °C for 4 h under rotary agitation, centrifuged at 2500 × $g$ for 3 min at 4 °C to discard the flowthrough and washed three times with 500 μL of IP buffer and one time with PBS only. The immune complex bound to the beads was eluted in 60 μL of Laemmli buffer 2X (Biorad cat.no. 1610747) incubated for 10 min at 95 °C, centrifuged at 2500 × $g$ for 3 min, and sent for MS analysis.

## Mass spectrometry from human cells

Liquid chromatography–tandem mass spectrometry was performed by the Proteomic Core Facility of EMBL (Heidelberg, Germany).

Sample preparation −Disulfide bridges were reduced in cysteine using dithiothreitol (56 °C, 30 min, 10 mM in 50 mM HEPES, pH 8.5) and such reduced cysteines were then alkylated with 2-chloroacetamide (room temperature, in the dark, 30 min, 20 mM in 50 mM HEPES, pH 8.5). Established protocols[67,68] were followed and trypsin (Promega, sequencing grade) was added (1/50 enzyme: protein ratio) for overnight digestion at 37 °C. The next day, the peptide recovery was performed by collecting supernatant on a magnet and combining it with a second elution wash of beads with HEPES buffer. Peptides were labeled with TMT6plex[69] Isobaric Label Reagent (ThermoFisher) according to the manufacturer's instructions. For the TMT6plex samples were combined and for the sample clean-up an OASIS® HLB μElution Plate (Waters) was used. Offline high pH reverse phase fractionation was carried out on an Agilent 1200 Infinity high-performance liquid chromatography system, equipped with a Gemini C18 column (3 μm, 110 Å, 100 × 1.0 mm, Phenomenex)[70].

LC-MS\MS − An UltiMate 3000 RSLC nano LC system (Dionex) fitted with a trapping cartridge (μ-Precolumn C18 PepMap 100, 5 μm, 300 μm i.d. x 5 mm, 100 Å) and an analytical column (nanoEase™ M/Z HSS T3 column 75 μm x 250 mm C18, 1.8 μm, 100 Å, Waters) were used. Trapping was carried out with a constant flow of trapping solution (0.05% trifluoroacetic acid in water) at 30 μL/min onto the trapping column for 6 min. Peptides were eluted by the analytical column with solvent A (0.1 % formic acid in water, 3 % DMSO) with a constant flow of 0.3 μL/min and an increasing percentage of solvent B (0.1% formic acid in acetonitrile, 3% DMSO). The outlet of the analytical column was coupled directly to an Orbitrap Fusion™ Lumos™ Tribrid™ Mass Spectrometer (Thermo) using the Nanospray Flex™ ion source in positive ion mode. The peptides were loaded into the Fusion Lumos via a Pico-Tip Emitter 360 μm OD x 20 μm ID; 10 μm tip (New Objective) and a spray voltage of 2.4 kV was applied with a capillary temperature of 275 °C. Full mass scan (MS1) was acquired with a mass range of 375-1500 m/z in profile mode in the orbitrap with 60000 resolution. The maximum filling time was 50 ms. Data-dependent acquisition (DDA) was performed with the resolution of the Orbitrap set to 15000, with a fill time of 54 ms and a limitation of 1×105 ions. A normalized collision energy of 36 was applied. MS2 data was acquired in profile mode.

IsobarQuant[71] and Mascot (v2.2.07) were used to process the acquired data, which was searched against a Uniprot Homo sapiens proteome database (UP000005640) containing common contaminants and reversed sequences. The following modifications were included in the search parameters: Carbamidomethyl (C) and TMT6(K) (fixed modification), Acetyl (Protein N-term), Oxidation (M) and TMT6 (N-term) (variable modifications). For the full scan (MS1) a mass error tolerance of 10 ppm and for MS/MS (MS2) spectra of 0.02 Da were set. Other parameters chosen: Trypsin as protease with an allowance of a maximum of two missed cleavages: a minimum peptide length of seven amino acids; and at least two unique peptides were required for protein identification. The false discovery rate on peptide and protein levels was set to 0.01. The raw output files of IsobarQuant (protein.txt - files) were processed using the R programming language (ISBN 3-900051-07-0). Only proteins that were quantified with at least two unique peptides were considered for the analysis. Raw signal-sums (signal_sum columns) were first cleaned for batch effects using limma[61] and further normalized using variance stabilization normalization[72]. Proteins were tested for differential expression using the limma package. The replicate information was added as a factor in the design matrix given as an argument to the 'lmFit' function of limma. Also, imputed values were given a weight of 0.05 in the 'lmFit' function. A protein was annotated as a hit with a false discovery rate (FDR) smaller than 5% and a fold-change of at least 2 and as a candidate with an FDR below 20% and a fold-change of at least 1.5.

## USP7 inhibition and in vivo cannula infusion

USP7 inhibitor: P5091 (Selleck Chemicals, S7132) was prepared freshly before injection.

P5091 was administered (10 mg/kg i.p. or 0.1 mg/kg intrathalamic) every day during the two-injection cocaine sensitization protocol (just before saline or cocaine administration (15 mg/kg) and at day 2–6) (Fig. 4).

For i.p. injection: the powder was first pre-dissolved in 5% v/v DMSO (vortex and sonication were used to aid dissolving and obtain a clear mix) then 95% v/v sesame oil (vortex was used to aid dissolving and obtain a clear mix) was added to the mix. Calculation was done to inject in i.p. a volume of 100 μL maximum and a dose of 10 mg/kg.

For cannula injections: the powder was first pre-dissolved in 54% v/v DMSO (vortex and sonication were used to aid dissolving and obtain a clear mix) then 26% v/v sesame oil (vortex and sonication were used to aid dissolving and obtain a clear mix) followed by 20% of artificial cerebrospinal fluid (125 mM NaCl, 2.5 mM KCl, 1.25 mM NaH2PO4, 25 mM NaHCO3, 1 mM MgCl2, 2 mM CaCl2, 25 mM D-glucose) was added to the mix. The calculation was done to inject through a cannula a volume of 400 nL and a concentration of 7.5 g/L.

We implanted 21-gauge guides that were cut 5.5 mm below a plastic pedestal (Plastics One Inc.) 0.2 mm above the PVT (from bregma, in mm): A-P −1.46, M-L + −1.02, D-V -2.8, angle: 10° and secured on the skull after H2O2 preparation using dental cement (Metabond kit, Parkell Inc.). For daily injection, an internal cannula of 26-gauge projecting 0.2 mm below the guide was inserted into the guide and connected to a 10 μL syringe (Hamilton). The injection rate was fixed at 0.1 μL/min. Contention was limited as the mice could freely move in their home cage during injection. No anesthetics were administered for daily injections.

## ChIPmentation

ChIPmentation experiments were performed based on the protocol of Schmidl C et al. [39] with minor modifications. The thalami of 2–4 WT or *Maged1* cKO mice injected either with cocaine or saline vehicle were microdissected and crosslinked in freshly prepared methanol-free 1% formaldehyde solution (Thermofisher #28906) for 10 min at room temperature. The crosslinking reaction was stopped with 0.125 M Glycine solution and the samples were washed 3x with PBS and stored at -80 °C. Samples were then resuspended in cell lysis buffer (10 mM Tris pH 8.0, 10 mM NaCl, 0.2% NP-40, 1X EDTA free Proteinase inhibitor cocktail; Sigma # 11836170001) and nuclei were released by homogenization with a pestle B –douncer, centrifuged 8 min at 600 g at 4 °C and resuspended in 130 μL of sonication buffer (50 mM Tris pH 8.0, 10 mM EDTA, 0.25% SDS, 1X Proteinase inhibitor cocktail). After 15 min at 4 °C the resuspended nuclei were transferred to sonication microtubes (Covaris # 520045) and sonicated in a Covaris S220 (to fragment chromatin to an average size of 200–400 bp). Sheared chromatin was centrifuged at 13,000 × g for 10 min to eliminate cell debris, and transferred to a new 1.5 mL tubes and diluted 2.5 times in ChIP dilution buffer (20 mM HEPES pH 7.3, 1 mM EDTA, 0.1% NP-40, 150 mM NaCl, 1X Proteinase inhibitor cocktail). H2AK119ub antibody–Dynabeads protein G complexes were added to the diluted chromatin and incubated overnight at 4 °C with gentle rotation. For each ChIP sample, 25 μL of Dynabeads protein G magnetic beads (Thermofisher #) were transferred to 1.5 mL tubes and washed twice with 1 mL of blocking buffer (1X PBS, 5 mg/mL BSA, 0.5% Tween 20, 1X Proteinase inhibitor cocktail), resuspended in 200 μL of blocking buffer and conjugated with 4 μg of H2AK119ub antibody (Cell Signaling, 8240 S) at 4 °C for 3 h in a rotating platform. Subsequently, antibody-bead complexes were washed twice and resuspended in 15 uL of blocking buffer and added to the diluted chromatin. For each wash, the beads were incubated for 5 min at 4 °C on a rotating platform, collected for 2 min in a magnetic stand, and resuspended in 1 mL of blocking buffer.

After overnight incubation chromatin–antibody-bead complexes were washed twice with 1 mL of RIPA/low salt buffer (10 mM Tris pH 8.0, 140 mM NaCl, 0,1%, 1 mM EDTA, 0.1 % Sodium Deoxicholate, 0.1% SDS, 0.1 % Triton X-100, 1X Proteinase inhibitor cocktail), twice with 1 mL of RIPA/high salt buffer (10 mM Tris pH 8.0, 500 mM NaCl, 0.1%, 1 mM EDTA, 0.1% Sodium Deoxicholate, 0.1% SDS, 0.1% Triton X-100, 1X Proteinase inhibitor cocktail), twice with 1 mlL of LiCl buffer (10 mM Tris pH 8.0, 250 mM LiCl, 1 mM EDTA, 0.5% Sodium Deoxicholate, 0.5% NP-40, 1X Proteinase inhibitor cocktail) and twice with 1 mL of 10 mM Tris pH8.0. Washes were performed as described for the antibody-bead conjugation. After the last wash beads were resuspended in 200 μL of 10 mM Tris pH8.0 and transferred to a new tube, collected with the magnet, and resuspended in 24 μL of Tagmentation Mix (1:1 Illumina 2x Tagmentation buffer: 10 mM Tris pH8.0). Resuspended chromatin-antibody complexes were incubated for 5 min at 37 °C and 1 μL of Illumina TDE1 enzyme was added to each sample. After mixing, each ChIP sample was tagmented for exactly 7 min at 37 °C, and the reaction was immediately stopped by the addition of 1 mL of RIPA/low salt buffer. Samples were then washed once more with 1 mL of RIPA/low salt buffer and once with 1 mL TE buffer (10 mM Tris pH 8.0, 1 mM EDTA). Finally, chromatin was eluted by resuspending beads in 100 μL of Elution buffer (10 mM Tris pH 8.0, 5 mM EDTA, 0.1% SDS, 0.3 M NaCl) supplemented with 50 μg of proteinase K and incubating at 65 °C overnight in an agitating thermoblock with heated lid. For total input chromatin samples, 10 ul of sonicated chromatin were diluted 10x in elution buffer with 50ugr of proteinase K and decrosslinked at 65 C overnight. In all cases, decrosslinked DNA was treated with 20ugr of RNAseA for 10 min at 37 C, and purified with the Qiagen minielute PCR purification kit (Qiagen # 28004). Approximately 10-20 ng of total input chromatin samples were then diluted in Illumina Tagmentation buffer and tagmented

with the TDE1 enzyme as for ChIP samples. Finally, DNA was re-purified with the Qiagen minielute PCR purification kit and size-selected with the Pronex paramagnetic beads.

In all cases, 2 μL of each sample was used for library quantification using the KAPA Library Quantification Kit (Roche #KK4923). The rest of the DNA eluate (20 μL) was used for library amplification using Nextera single-index primers as previously described using the KAPA HiFi hot-start ready mix (Roche # KK2601). The number of cycles used to amplify each library was determined as the Ct value of the qPCR reaction +1. For library quantification and amplification, the KAPA HiFi polymerase was preactivated by heating at 98 °C for 30 s. In all cases, a nick repair step was performed before PCR amplification by incubating the PCR reactions at 72 °C for 6 min. PCR-amplified libraries were purified and size selected for fragments of 200-250 bp using the Pronex paramagnetic beads (Promega # NG2001) according to manufacturer instructions. Finally, libraries were quantified using the Qubit High Sensitivity DNA quantification kit (ThermoFisher # Q32851) and analyzed by Fragment analyzer and sequenced pair paired-end sequencing 100 bp in a MGISEQ-2000 platform (BGI sequencing services, Honk Kong).

## H2Aub enrichment analysis

ChIPmentation sequencing data were analyzed using the public Galaxy server (https://usegalaxy.eu)[73] and personalized R scripts. First, adapter sequences were trimmed from both R1 and R2 read files using Cutadpat v3[74] and mapped against the mouse mm10 genome assembly using Bowtie2 (v 2.3.4.1)[75]. Significantly enriched regions were called using MACS2 (v 2.1.1.20160309.3) with broad peak calling parameters[76]. The bam files from the total input chromatin samples were used as a control for MACS peak calling. Coverage files were normalized by the number of millions of valid paired reads of each sample. The R (v4.1.2) and Rstudio (1.4.1106) software were used for downstream analysis.

For the analysis of H2Aub+ elements, only MACS2 peaks that were in common in at least two biological replicates were selected for further analysis using the R GenomicRanges package (v1.44.0)[77]. The distance distribution of peaks from the nearest TSS was calculated using the chipenrich package[78]. The mouse GCR38 gene annotation was obtained using the R annotationHub package (v3.0.2) and used to select gene promoters (defined as the genomic region 5 kb upstream and 2 kb downstream of the gene TSS) overlapping with H2Aub peaks. Differentially enriched H2Aub+ genes were thus defined as those loci that displayed H2Aub+ promoters between two experimental conditions. We also identified differential H2A enriched genes by comparing H2Aub+ elements called in saline vs cocaine-injected mice and attributing these peaks to genes using the chipenrich package (polyenrich method). The conclusions of this analysis were overall similar to those reached based on the analysis of H2Aub promoters (data not shown). However, we preferred to base subsequent analyses on the H2Aub+ promoter because of the limitations of long-range enhancer assignment to target genes. R scripts used for H2Aub peak analysis are publicly available at GitHub[79].

The H2Aub antibody used in this study has been used in ChIPseq experiments in several previous works (e.g. Kallin et al., 2009[80]). The genome-wide distribution of H2Aub enriched regions closely parallels that of other studies[80,81] with a high enrichment at gene promoters (Supplementary Fig. 15). Besides, this distribution differs from the ubiquitous and promoter-enriched distribution described for H2A (e.g, Zhang et al. 2005[82]), supporting that our H2Aub antibody specifically recognizes this epigenetic modification rather than non-specifically immunoprecipitating H2A-bound chromatin. As shown in Supplementary Fig. 15, H2A is broadly distributed genome-wide with only mild enrichment over gene bodies and is depleted from the nucleosome-free regions of gene transcription start and transcription termination sites. Instead, H2Aub is mostly enriched at gene

 

promoters. Thus, the differences in our H2Aub coverage profiles with those reported for H2A and their resemblance to the H2A distribution of Yang et al., 2014[81] strongly suggest that the H2Aub enrichments reported in our study are specific.

## Gene ontology and curated gene set analysis

Gene Ontology analyses of biological processes of genes displaying either altered H2Aub binding in their promoters and/ or that were differentially expressed across the different experimental conditions were performed using the gprofiler2 (v0.2.1), ClusterProfiler (v4.0.5) and enrichplot (v1.12.3) packages[83,84]. For differentially expressed gene sets, gprofiler analysis was performed by ranking genes according to their FC value, while the unranked gprofiler test was used for gene sets displaying differential H2Aub binding in their promoters or for gene lists resulting from the crossing of RNAseq/ChIPmentation data analysis. Finally, the GSEA software[85] was used for curated gene set analysis of differentially expressed genes. R script used for gene onmtology analysis are available at GitHub[86].

## Statistics and reproducibility

Statistical analyses were performed using GraphPad Prism 9 software (GraphPad Software Inc.) and SPSS Statistics 27 (IBM). Parametric data, provided by the Kolmogorov–Smirnov test, were analyzed using a Student's $t$-test, one-way and two-way ANOVA tests (when a value was missing (because of a random reason), we performed a mixed-effects model instead of a two-way ANOVA) and Tukey's or Sidak's post hoc tests when appropriate, as indicated in figure legend. For non-parametric data, the Wilcoxon Mann-Whitney test for two-sample comparisons or the Kruskal–Wallis test for multiple comparisons was used. Differences were considered significant when $P < 0.05$. The results are expressed as mean ± standard error of the mean (s.e.m.). The threshold for significance was set at $P < 0.05$. Sample sizes ("n") are given as the number of animals used for experiments.

For RNA-seq statistical analysis we used the 3D RNA-seq. To establish a significant differential list of genes, an FDR < 0.05 was chosen and a log2 Fold-change threshold > 0.5 or <0.5 was applied when mentioned.

The representative western blots shown in Fig. 3g, h, k were repeated at least 3 times with similar results, and 2 times for Fig. 3i.

## Human genotyping, sample selection and clinical assessment

Men and women > 18 years seeking treatment for any SUD other than nicotine were consecutively recruited between April 2008 and July 2016 through two multicentric protocols. Both protocols and the current study were approved by the relevant Institutional Review Boards [CPP Ile-de-France IV and CEEI from the Institut de la Santé et de la Recherche Médicale (INSERM), IRB00003888 in July 2015, respectively]. The clinical trial registration numbers are NCT00894452 and NCT01569347. All participants provided written informed consent for both the clinical and genetic assessments, and study records were continuously monitored by the hospital research administration. The research was conducted in accordance with the Helsinki Declaration as revised in 1989. Eligible participants were excluded if they had severe cognitive impairment or insufficient mastery of the French language preventing misunderstanding of the study purposes and assessments, if they had no social insurance, and if they were under compulsory treatment or mentorship. They did not receive compensation for participating in the study.

Assessments included history of substance dependence (the syndrome equivalent to severe substance use disorder in the DSM 5 and revealing the presence of addiction) using the E section of the Structured Clinical Interview for DSM-IV (SCID-IV)[87] and socio-demographic elements (currently married, yes vs. no; currently employed at least 20% time job, yes vs. no, ever experienced home-lessness – defined as having spent at least three months in the streets,

yes vs. no). For the current study, we focused on cocaine-related variables that were modeled in the preclinical study, namely: lifetime cocaine dependence (presence/absence), age at onset (AAO) of first cocaine use, and time from first cocaine use to cocaine dependence (hereafter termed "TRANSCOC") - considered to be related to the rewarding properties of cocaine, which were modeled in the preclinical part - and the behavioral components of the Scale for Assessment of Positive Symptoms – Cocaine-Induced Psychosis (SAPS-CIP) considering total score, and aggression and stereotypies subscores[88].

**Biological sampling and genetic analyses.** DNA was extracted from whole blood using a Maxwell 16 PROMEGA® extractor (Promega France, Charbonnières-les-Bains, France). Purity assessment followed the procedures described by the Center National de Recherche en Génomique Humaine. Participants were genotyped using the Infinium PsychArray (Illumina, San Diego, CA, USA) in two batches (2015 and 2017) by Integragen SA® (Evry, France) using the same pipeline. Genotype files were merged for the present study, keeping only bi-allelic variants common to both extractions. Then, PLINK 2.0[89] was used for quality control (QC), based on a consensus procedure for ancestry, relatedness, and genetic discrepancies[90], followed by imputation on the Michigan Imputation Server v1.0.4 using the reference file hrc.r1.1.2016 through a SHAPEIT + MiniMac3 pipeline[91]. The number of post-QC variants increased from 552,000 to 7,833,448 SNPs. Chromosome X was imputed separately by Impute2 after pre-phasing by Shapeit4, following a consensus procedure[92], after the exclusion of the pseudo-autosomal region. This increased the number of post-QC variants from 12,942 to 34,243 on this chromosome.

Quality control and ancestry assessment were performed on a whole-genome level using a standard procedure[90] based on: an overall call rate > 99%, missing genotypes per study and individual <5%, Hardy-Weinberg equilibrium held at $P > 10^{-6}$, and relatedness inferior to the 2nd degree (in case two participants fell within such relatedness, the one with the highest overall call rate was kept for the study). From 592 genotyped patients, 522 were available for analysis, 351 of whom had strict Caucasian ancestry as compared to the distribution of variants in Europeans from the 1000 genomes database.

The candidate gene study was performed using genomic positions (human genome version GChR37) of the genes prioritized by the pre-clinical study, keeping markers with minor allele frequency (MAF) > 5 %, namely: *MAGED1* (chromosome X, 72 markers), *USP7* (chromosome 16, 142 markers), and Histone 2 A genes (chromosome 1, *HISTH2AC18* and *HISTH2AC20*, no marker). Variants showing significant associations with any of the phenotypes of interest were annotated for their biological function and plausible impact according to multiple knowledge-based repositories online, as previously suggested[93]: the Combined Annotation-Dependent Depletion (CADD) database[94], brain expression and methylation quantitative trait loci (eQTL through BRAINEAC, http://www.braineac.org/, and mQTL through the mQTLdb database, http://www.mqtldb.org), and their ability to bind DNA enzymes and/or modify DNA conformation (regulomeDB database http://www.regulomedb.org). We also considered linkage disequilibrium (LD, using the $r^2$ measure), which indicates how minor alleles tend to be inherited altogether due to chromatin blocks. High LD (considered at $r^2 > 0.9$ in our study, based on European reference panels) indicates a similar genomic signal from possibly multiple SNPs, among which both the strongest statistical association and the highest functional impact can be used to identify the one(s) responsible for the biological signal behind the statistical association. Additionally, we identified CNVs following a pipeline[95] based on the Illumina® final report file generated by GenomeStudio v.2, which involves two programs, PennCNV[96] and QuantiSNP[97], in a combined algorithm. Both programs are based on the hidden Markov chain model but show different sensitivity and specificity. CNVs may be very rare but represent the closest variation to gene deletion in rodent models.

**Human SNP-based data analysis.** Continuous variables were described by means (standard deviation, SD) or medians (interquartile range, IQR) depending on their distribution, and qualitative variables are described by absolute counts and frequencies. Nonparametric tests were used to identify the clinical and sociodemographic factors associated with cocaine-related variables.

SNP-based analyses were performed using PLINK linear regression adjusted for sex and the first ten principal components of ancestry, a stringent adjustment recommended to further minimize the risk of population stratification. Possibly censored variables related to time, namely: AAO of cocaine dependence and transition from first cocaine use to CUD, which were analyzed using Cox proportional regression models in R. All these analyses included age, biological sex, and the first ten ancestry components as covariates and cofactor. CNVs in the candidate genes -if any- were also tested for associations with cocaine-related variables based on frequency, size, or type (duplication *vs.* deletion) in the subsample of Caucasian ancestry. Statistical significance was set at P < 0.05 after correction for multiple testing using a false discovery rate procedure (Benjamini-Hochberg, BH-FDR). Separately for chromosomes X and 16, the *p* values of all seven phenotypes and SNPs were listed together to yield the corresponding FDR. In the case of multiple statistical associations (which is expected when using imputation), we used linkage disequilibrium (LD, measured by r2) and in silico gene expression measures in the brain (http://www.braineac.org/, Supplementary Data 3) to identify lead SNPs as the most plausibly associated with the underlying biological signal. These lead SNPs (identified as having LD $r^2 < 0.9$, strong statistical association, and reliable evidence for functional impact) were further characterized using graphical representations and adjustment for more sociodemographic and clinical confounders, depending on the phenotype they were associated with. Analyses were conducted with R 4.02 through R studio 1.4.1103 under Mac OS X.12.6. The current study follows the STREGA guidelines for the report of genetic studies[86].

The summary of the R session and the summary statistics are available in Supplementary Data 4. Of note, no QQ-plots were produced given the within-cases, candidate-gene design of the study, driven by preclinical experiments.

### Reporting summary

Further information on research design is available in the Nature Portfolio Reporting Summary linked to this article.

## Data availability

The data supporting the findings are available within the article and its Supplementary Materials and are available from the corresponding author upon request. All RNA-seq and ChIPmentation data are deposited in GEO under accession number GSE208142. The mass spectrometry data is provided in Supplementary Dara 2. The R scripts used for this study are available upon request. Source data are provided with this paper.

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

## Acknowledgements

We thank Dr. Christian Luscher, Dr. Antoine Adamantidis, Dr. Ami Citri, Dr. Marina Picciotto, Dr. Gilbert Vassart, and Dr. Michel George for their critical reading and their helpful insights on our manuscript. We thank Delphine Houtteman, Souad Laghmari, and Laetitia Cuvelier for technical assistance and mice colonies in Brussels. We thank Sarah Gharbi for technical assistance in Grenoble. We thank the EMBL Proteomics Core Facility. JC is a research fellow of FRS-FNRS and Fonds Erasme. LB is a Ramon y Cajal fellow (CBMSO). AKE is a research director of the FRS-FNRS and a WELBIO investigator. Funding was provided by FRS-FNRS grants 23587797, 33659288, 33659296 (A.K.E.), WELBIO grant 30256053 (A.K.E.), Fondation Simone et Pierre Clerdent 2018 Prize (A.K.E.), Fondation ULB (A.K.E.), Fondation Cigrang (A.K.E.), Dotation Jeune Chercheur INSERM (L.B.), AFM stratégique 2 MyoNeurAlp (L.B.), Agence Nationale pour la Recherche grant ANR- 21-CE11-0013 (S.B.), EMBL Interdisciplinary Postdocs (EIPOD4) initiative co-funded by Marie Skłodowska-Curie (grant 847543) (T.C.), Mission Inter-Ministérielle de Lutte Contre la Drogue et la Toxicomanie grant ASE07082KSA (F.V.), Direction de la Recherche Clinique et du Développement de l'Assistance Publique – Hôpitaux de Paris grant OST07013 (F.V.), French Ministry of Health, Programme Hospitalier de Recherche Clinique National AOM10165 (F.V.).

## Author contributions

Conceptualization: J.C., A.K.E.; Investigation: J.C., L.B., P.H., R.I., T.C., C.D., R.I., F.V., J.B., E.M., R.C.; Visualization: J.C., L.B., P.H., R.I., T.C., F.V., J.B., A.K.E.; Data Analysis: J.C., L.B., R.I., M.D., T.C., S.B., F.V., E.M., R.M., A.K.E.; Funding acquisition: S.B., F.V., A.K.E.; Project administration: J.C., A.K.E.; Supervision: A.K.E.; Writing of original draft: J.C., A.K.E.; Writing, review and editing: J.C., L.B., R.I., M.D., T.C., S.B., F.V., J.B., A.K.E.

## Competing interests

The authors declare no competing interests.
