## [Peer Review File · Nature Communications]

USP7/Maged1-Mediated H2A Monoubiquitination in the Paraventricular Thalamus: An Epigenetic Mechanism Involved in Cocaine Use DisorderREVIEWER COMMENTS

Reviewer #1 (Remarks to the Author):

This is a very interesting study by Cheron and colleagues in which they demonstrate a role for Maged1 in cocaine sensitization and cocaine-induced transcriptomic changes, leading to interacting proteins like histone H2A which was disrupted in Maged1-cKO samples and displayed a differential association with Maged1 in cocaine vs saline treated mice. Mage proteins associate with E2/3 ubiquitin ligases, and indeed H2A becomes monoubiquitinated. The work is entirely focused on experimenter administered cocaine, using a chronic cocaine protocol typical of the field, and analysis of sensitization behavior. The study concludes with examination of potential SNPs in Maged1 from human patients with CUD. Overall, the study makes an important contribution to our understanding of the molecular mechanisms of cocaine action, the mechanisms engaging a H2A monoubiquitination (involved in gene repression as demonstrated in past studies in the field), and understanding Maged1 function. The main limitation is the experimenter delivered cocaine. It would really strengthen the study to demonstrate a role for Maged1 and H2AUb in cocaine IVSA for example, as it has much more face validity. The other main concern is sex as a biological variable. Specific comments are below.

1) Would be good to show that locomotor performance isn't itself impaired in vGluT2-Cre::Maged1 loxP mice, or Maged1loxP-AAV-Cre mice (related to figure 1 sensitization data). There seems to be an initial distance traveled difference between Maged1KO and control, which is also observed in the vGluT2-Cre::Maged1loxP and control. Thus, it appears that the knockout of Maged1 affects locomotor performance by itself, even in the absence of cocaine, which then confounds interpretation of the sensitization data.

2) Experimenter administered chronic cocaine (for Figure 2) similar to sensitization experiment (in Figure 1), but would be ideal to see what happens with cocaine IVSA, a more meaningful assessment of cocaine seeking and taking.

3) Is it correct that the Maged1 IP brought down H2A only in the presence of cocaine, where as the H2AUb IP brought down Maged1 in both saline and cocaine? Does the Maged1 IP bring down H2AUb?

4) In Figure 3g, h, the distance traveled is much less than what is shown in Figure 1 for sensitization. Why is that? The locomotor behavior across all days should be shown as well.

5) Do the patients included in Figure 4 have a sole history of cocaine use? Or, more likely, do they also report other drugs of abuse (alcohol, nicotine, etc) or prescription drugs? It is important to share the complete drug history, not just cocaine. Were only men included in this analysis? If so, why?

6) It was not entirely clear if both males and females are being used. This is clearly an important issue with regard to sex as a biological variable, and cocaine demonstrating well known sex-specific effects in mice and humans. And Maged1 is an X-linked gene, so males and females should be examined.

Reviewer #2 (Remarks to the Author):

Cheron et al report a comprehensive analysis of the regulation of H2AUb in the mouse thalamus following cocaine treatment. The examination of histone ubiquitination in the context of cocaine exposure is a novel and valuable contribution to the field of neuroepigenetics. The manuscript is important in that characterization of H2AUb in this context is timely and highly rational, given that it is known to be regulated by cocaine in several brain reward regions. The manuscript includes a plethora of approaches to comprehensively interrogate H2AUb within the context of a putative H2AUb regulatory protein, Maged1, using a Maged1 conditional knock-out mouse line. These

include cocaine locomotor sensitization behavior, RNA-seq, ChIPmentation, co-IP MS/MS, co-IP western blotting, and human genomics. The conclusions are extremely compelling and exciting, namely, (1) rigorous and innovative experimental approach for conditional knockout of Maged1; rationale for the PVT region of knockout is strong (2) comprehensive and robust analysis of the effect of Maged1 across multiple brain regions, including the robust attenuation of cocaine locomotor sensitization with glutamatergic neuron Maged1 cKO in PVT, (3) report an increase in H2Aub in several brain regions, (4) ChIP-sequencing of H2Aub in thalamus following cocaine treatment and (4) identification of human MAGED1 variants that associated with cocaine use disorder.

Enthusiasm for this manuscript is dampened by several major concerns, which are broadly listed below. Detailed feedback and requests for information and revisions are found in the accompanying annotated PDF.

- It is unclear if the Maged1-cKO expresses a truncated Maged1. This is suggested by the the co-IP MS, for which the authors report a loss of only one co-IP'd protein between wild-type and Maged1-cKO condition. The expression of a truncated Maged1 protein in the cKO condition would require a major revision to the manuscript, with respect to (1) validation of the truncation and (2) conclusions drawn.
- Conclusions about the relative expression of Maged1 in PVT are vague and derive from unquantified, 'qualitative' imaging analyses.
- The organization of the manuscript leads to some confusion (eg the abstract and results are not well-aligned; details of the truncation appear only in the second section although the cKO appears in the first section)
- The results sections do not include pertinent details that are necessary to interpret the findings (eg cocaine dose and treatment paradigm, sex and age of subjects)
- Female mice are excluded without a rationale provided
- An additional expert on human genomics must necessarily review the CUD association study methods and results

Reviewer #3 (Remarks to the Author):

Review for Nat.Comm. 2023: Histone H2A monoubiquitination in the thalamus regulates cocaine effects and addiction risk

The end goal of this study is to provide new risk factors and therapeutic targets for cocaine use disorder (CUD). In a previous study the authors identified Maged1 as a master regulator of cocaine reward and reinforcement in mice. Therefore, they centered this work on the characterization of the molecular mechanisms involving Maged1 in these cocaine-related phenotypes. The authors claim that locomotor sensitization in response to chronic cocaine usage requires the expression of Maged1, specifically in vGluT2 neurons and particularly in the paraventricular thalamus (PVT). Furthermore, Maged1 inactivation mainly led to an altered transcriptional regulation in the thalamus after cocaine administration. Through IP-MS experiments two of Maged1 partners were validated, namely: USP7 and H2A, the latter exclusively upon cocaine administration. Using human cell lines, the authors further confirmed the conserved nature of Maged1 and USP7 binding. Importantly, USP7 inhibitor recapitulated the loss of locomotor sensitization and was accompanied by an increase in H2Aub baseline levels. Interestingly, H2Aub thalamic ChIP-sequencing showed that cocaine administration in control mice increased H2Aub peaks and that genes displaying H2Aub cocaine-enrichment did not show enrichment in Maged1-cKO animals, stressing the link between cocaine dependent H2Aub gene regulation and Maged1 mechanism of action. This manuscript describes the different roles of Maged1 in different regions in the context of cocaine usage and for the first time its function in the PVT neurons has been addressed. The authors provide new sequencing datasets that support the hypothesized link between transcriptional repression observed as a consequence of cocaine administration, and H2Aub epigenetic control of such transcriptomic effect. While this work correlates H2Aub genomic regulation to CUD, it does not causally address its role in the development or maintenance of its behavioral feature of locomotor sensitization. Thus, already the title is misleading. The authors

most of the time tend to overstate their conclusions and relate them to the epigenetic regulation, however most of the times their findings mostly focus on Maged1 and USP7 mechanisms of action. We advise for a cautious rephrasing of the key statements of this manuscript and the following major and minor revisions.

Major

Fig. 1

In the text it is written that (73-77 lines) "To screen for potential vGluT2 nuclei involved in acquisition of this phenotype, we performed AAV retro mCherry injection in the NAc and revealed the amygdala, the ventral subiculum and the paraventricular thalamus (PVT) as the main vGluT2 nuclei targeting the NAc (Fig. 1g,h). Of these, Maged1 mRNA in situ hybridization and immunohistochemistry (IHC) showed a singularly high level of Maged1 expression (Fig. 1i) in the PVT, which has recently been shown to be involved in drug addiction." From the results reported, we only see a zoomed in image of PVT Maged1 expression, but there are no quantifications validating the claim of higher expression of this protein in the PVT compared to the other regions. Additionally, from Extended Data Fig.1 it seems that Maged1 is ubiquitously expressed across the brain rather than a singularly high concentration in the PVT nucleus. Either rephrase or provide quantifications.

Extended Data Fig.3/4

Panel d shows that the abolishment of locomotor sensitization found in the full Maged1 KO mice model was recapitulated by either the conditional-KO of Maged1 in vGluT2 or PVT neurons, but not other brain nuclei (see Ext. Fig.3, panel d). This finding is partially in contrast with their previous discovery where the cKO of Maged1 in the mPFC led to an impaired phenotype using the same behavioral protocol (De Backer JF et al., EMBO Rep, 2018). The difference in these results has not been addressed in this study. Moreover, from the PFC and AMG re-expression experiment reported in Ext. Fig.4 (panel a and b) it seems that Maged1 rescue in a full KO model partially restores the cocaine sensitization behavior. To conclude that the PVT is the key nucleus necessary and sufficient for this phenotype a direct comparison to the Fig.1 panel m dataset would be required.

Fig. 2

Adjust Fig.2 panel a to more clearly describe the mice line used in this experiment. There is no information regarding the mouse line used to evaluate the efficiency of the FACS sorting procedure. Moreover, since for the RNA-sequencing experiment no FACS sorting was used, it is unclear which mouse line was processed. Additionally, is there any marker used to assess that the manual dissociation of the thalamus was precisely leading to a clean PVT isolation?

Panel c-d related: These graphical representations do not back up the authors claim "Interestingly, a gene set enrichment analysis with curated gene sets revealed that cocaine-regulated genes in control but not in Maged1-cKO mice were targets of the Polycomb repressive complex (PRC) (Fig. 2c,d and Extended Data Table 1). Notably, the genes with expression similarly regulated by cocaine in the control and Maged1-cKO mice were not targeted by PRC action." While from the curated gene sets plot it is possible to deduce that around one third of the downregulated genes after cocaine treatment are part of the PRC2 targets in control conditions, the same information is not reported for Maged1 cKO mice. Additionally, the bottom illustration should include information concerning the expression levels related to each of the proteins depicted and how this value changes across experimental conditions; thus better highlighting the involvement in Maged1 with PRC key players.

The change in GO hits reported in Extended Data Fig.6 is not enough to describe which pathways are the most affected in the cKO model. Similar to Ext. Data Fig.5 panel d, a GO for cKO mice should be included.

Panel g-i related: To test whether Maged1 interaction with H2A and USP7 contributes to the regulation of the monoubiquitylation of H2A at lysine 119 (H2Aub) the authors use IP experiments and conclude that "these data suggest that Maged1 plays a role in H2A monoubiquitination in thalamic neurons following cocaine intake, via Maged1 involvement with a PRC1 member". This interpretation is an overstatement as what the experiment shows is that the binding between USP7 and H2Aub depends on the presence of Maged1, however, per se the H2Aub PTM is present with or without Maged1 as confirmed in Fig.2h. Importantly, as Fig.2h and 1 refer to saline conditions, no conclusion can be drawn about cocaine effect on such PTMs, but only confirm that the binding between H2Aub, USP7 and Maged1 is not altered by cocaine consumption (see Extended Data Fig.8).

Furthermore, it is not clear what the starting material for the Co-IP experiment was.

Extended Data Fig. 7

To reinforce the author's claim "we observed that Maged1 and USP7 protein levels were both higher in the PVT than in other parts of the thalamus", a thorough quantification of Maged1 and USP7 in all thalamic nuclei should be included.

Fig. 3

Panel b-e: from the representative WB view reported in panel b, it seems that H2A has a high variability across conditions and therefore cannot be used as a normalizer in the following analysis (c-e). Compare the results obtained with a B-actin normalization and/or an additional protein known not to be subjective to Maged1 and cocaine manipulations. Please include (as supplementary) the representative WB images relative to NAc and PFC as well. Could an IHC for H2Aub specifically directed at the vGlut2 PVT neuronal population be added in support of this findings?

Line 177

"These results showed that pharmacological abolition of cocaine-evoked H2Aub deposition in the thalamus impeded cocaine-evoked behavioral sensitization." This conclusion should be reconsidered as it stems from the USP7 pharmacological inhibition. Therefore, by blocking USP7 deubiquitinase activity, the sensitized behavior is altered. The result not mentioned but worth discussing, is that saline treated mice have a higher H2Aub levels. Could it be possible that a further increase due to cocaine administration is contrasted due to ceiling effects? Additionally, what could be the biological impact of such a higher baseline in H2Aub levels? In essence, any strong implication of H2Aub deposition and their causal role in the cocaine evoked phenotype is an overstatement as this could be only hypothesized due to the correlation between the two phenomena.

Finally, how specific is the USP7 inhibitor? There is no evidence for its specificity and important controls are missing.

Human data:

Has the patient cohort been controlled for other types of medication?

Minor:

Extended Data Fig.4: report group size for experiment panel d.

Fig.2: panel e, add on top of each column the IP antibody to help the reader navigate the WB.

Fig.2: panel f, the order of the volcano plots and the corresponding legend descriptions are inverted. The legend used on the upper part of the volcano plots are misleading, please correct. (the left and right ends of the plots do not correspond to the groups highlighted with the labels but they reflect a higher or lower expression compared to the reference group used).

Fig.2: panel l-m same as before, legend used on the upper part of the volcano plots are misleading.

Line 144 add Fig number.

Fig.3 panel K: What has been used to normalize H2Aub values? Any representative images?

Fig.3 panel O: wrong control labelling.

Line 246, discussion section, the word "complete" is inappropriate especially when referring to the restoration experiment where only a partial rescue of cocaine sensitization was obtained.

Responses to reviewers.

Reviewer #1 (Remarks to the Author):

This is a very interesting study by Cheron and colleagues in which they demonstrate a role for Maged1 in cocaine sensitization and cocaine-induced transcriptomic changes, leading to interacting proteins like histone H2A which was disrupted in Maged1-cKO samples and displayed a differential association with Maged1 in cocaine vs saline treated mice. Mage proteins associate with E2/3 ubiquitin ligases, and indeed H2A becomes monoubiquitinated. The work is entirely focused on experimenter administered cocaine, using a chronic cocaine protocol typical of the field, and analysis of sensitization behavior. The study concludes with examination of potential SNPs in Maged1 from human patients with CUD. Overall, the study makes an important contribution to our understanding of the molecular mechanisms of cocaine action, the mechanisms engaging a H2A monoubiquitination (involved in gene repression as demonstrated in past studies in the field), and understanding Maged1 function. The main limitation is the experimenter delivered cocaine. It would really strengthen the study to demonstrate a role for Maged1 and H2AUb in cocaine IVSA for example, as it has much more face validity. The other main concern is sex as a biological variable. Specific comments are below.

We sincerely appreciate the reviewer's thoughtful assessment, acknowledging our study's significant findings on Maged1's role in cocaine adaptive-behaviors, transcriptomic changes, and H2A monoubiquitination, that is new to the field, while considering the valuable suggestions for additional studies involving self-administered cocaine.

1) Would be good to show that locomotor performance isn't itself impaired in vGluT2-Cre::Maged1 loxP mice, or Maged1loxP-AAV-Cre mice (related to figure 1 sensitization data). There seems to be an initial distance traveled difference between Maged1KO and control, which is also observed in the vGluT2-Cre::Maged1loxP and control. Thus, it appears that the knockout of Maged1 affects locomotor performance by itself, even in the absence of cocaine, which then confounds interpretation of the sensitization data.

Indeed, spontaneous locomotor activity is decreased in Maged1 KO mice (De Backer et al. 2018) as well as in vGluT2-Cre::Maged1 loxP mice, or Maged1loxP-AAV-Cre mice in the PVT. To address this issue, we decided to normalize the distance traveled during a two-injection cocaine sensitization experiment on saline day 0. Using this normalization method, we observed that the acute response to the first injection of cocaine was similar between the vGluT2-Cre::Maged1 loxP mice and the control mice, however, we observed a specific sensitization abolition in the cKO mice, contrasting with the control mice (Extended Data Fig. 10a).

To further dissociate a global effect on locomotor activity rather than a specific effect on cocaine-induced sensitization of Maged1 in the PVT, we compared the locomotor activity on the first day of saline injection and the last day of cocaine injection (after reaching the ceiling level) in the rescue model, Maged1 KO, AAV-Maged1 (PVT), and in the cKO model Maged1 loxP, AAV-Cre (PVT). While there was no spontaneous locomotor activity difference between the inactivation or the re-expression of Maged1 in the PVT in these 2 models, the sensitization was significantly stronger in the rescue group as illustrated by an interaction effect between mice group (rescue versus deletion in the PVT) and treatment (saline/cocaine) (Extended Data Fig. 13). This analysis and discussion has been added in the supplementary text.

2) Experimenter administered chronic cocaine (for Figure 2) similar to sensitization experiment (in Figure 1), but would be ideal to see what happens with cocaine IVSA, a more meaningful assessment of cocaine seeking and taking.

We greatly value your invaluable feedback and suggestions, including your proposition to further investigate the impact of Maged1 on cocaine self-administration using intravenous self-administration (IVSA) techniques. In direct response to your insightful recommendation, we executed an IVSA experiment involving vGluT2-cre::Maged1lox mice alongside wild-type (wt) mice. Additionally, we carried out a Cocaine Conditioned Place Preference (CPP) paradigm on additional groups of mice.

During the training phases of fixed ratio 1 (FR1) and FR3 sessions in ISVA, we did not observe significant differences in active nose-poking responses between vGluT2cre Maged1::lox mice and WT controls (Extended Data Fig. 3). However, statistical significance emerged during the subsequent acquisition phase, spanning three consecutive days and satisfying criteria of stability, discrimination, and more than ten infusions. Notably, the number of active nose-poking responses displayed a marked increase in vGluT2cre::Maged1lox mice compared to WT controls (Extended Data Fig. 3d). This finding suggests that Maged1 in vGluT2 neurons may play a role in modulating the initiation of cocaine self-administration behaviors. We have included this new results in the main text and in the supplementary material. To assess the reinforcing properties of cocaine from a different perspective, we conducted a CPP experiment. The results from our CPP experiment revealed a significant distinction between the two experimental groups. Notably, vGluT2-mediated Maged1 inactivation led to a reduction in the time spent in the cocaine-paired compartment (Extended Data Fig. 3a). This alteration aligns with the effects observed in Maged1 KO mice, contrasting with the outcomes observed in the previously targeted regions as reported in our earlier study (De Backer et al. 2018).

These results indicate the early involvement in cocaine adaptive behaviors and are in accordance with previous results shown in this study. While this significant result provides a valuable insight into the potential impact of Maged1 in vGluT2 neurons, we acknowledge the wider complexities of this finding.

Indeed, as a follow-up to this study, we are investigating the mechanisms underlying this effect and going beyond this acquisition phase. To further investigate the involvement of this novel epigenetic mechanisms in vulnerability to drug addiction, we have initiated a new study using optogenetic self-stimulation of VTA DA neurons (oDASS) (Pascoli et al. 2015). Indeed, in line with our human data (altered transition time to addiction), using oDASS, we aim to decipher the mechanisms that distinguish mice that switch to compulsive behaviors from those that abstain when faced with punishment (Pascoli et al. 2018). oDASS will also be combined with the characterization of post-synaptic NAc neurons activity by calcium imaging (Varin et al. 2023) to better understand the role of Maged1 in the PVT in this context. Furthermore, these innovative approaches have the potential to be extrapolated to encompass various drugs of abuse by reducing their off-target effects. We believe that including these consequential experiments goes beyond the scope of this article and warrants a separate study, primarily due to the substantial time they necessitate.

3) Is it correct that the Maged1 IP brought down H2A only in the presence of cocaine, where as the H2Aub IP brought down Maged1 in both saline and cocaine? Does the Maged1 IP bring down H2Aub?

Yes, it is correct to say that Maged1 IP brought down H2A only in the presence of cocaine as it was shown by our IP-MS and IP-WB results. It is also correct to say that H2Aub brought down Maged1 in both saline and cocaine conditions. However, with our working antibodies, we cannot directly confirm that Maged1 IP bring down H2Aub as we found a light IgG chain

contamination at the expected size for H2Aub band (Maged1 and H2Aub are both rabbit antibodies). This served as a motivating factor for co-immunoprecipitating H2Aub and looked for Maged1. Furthermore, USP7 IP data converge toward an interaction between H2A(ub) and Maged1. Our interpretation is that the interaction between H2A(ub) and Maged1 exists in both conditions (saline and cocaine) but that cocaine triggers this interaction making the detection easier (Extended Data Fig. 12).

4) In Figure 3g, h, the distance traveled is much less than what is shown in Figure 1 for sensitization. Why is that? The locomotor behavior across all days should be shown as well.

For this experiment, involving a preclinical drug injection (P5091), we decided to slightly modify the classical sensitization protocol and we used the two-injection protocol as described in Valjent et al. 2010. There are two main reasons for that: (1) the sensitization effect is demonstrated to be higher after a week interruption between the 2 injections of cocaine – potentially increasing the sensitivity for an effect of a preclinical drug (Valjent et al. 2010) and (2) it allowed a longer time period between the 2 cocaine injections to deliver a preclinical drug lowering the risk of pharmacological interaction. In this two-injections protocol we recorded the locomotor activity for 30 minutes (Fig. 3) instead of the classical 60 minutes (Fig. 1).

5) Do the patients included in Figure 4 have a sole history of cocaine use? Or, more likely, do they also report other drugs of abuse (alcohol, nicotine, etc) or prescription drugs? It is important to share the complete drug history, not just cocaine. Were only men included in this analysis? If so, why?

The clinical sample was 23% female, as described PL, and as is commonly seen in clinical samples of severe SUD patients (see, e.g., Palma-Álvarez et al. 2023). We acknowledge that the manuscript lacked a complete description of the clinical sample.

First, we have modified the sentence describing the sample, PL, “with 350 consecutively recruited outpatients with cocaine addiction and genetically verified Caucasian ancestry. After demonstrating the absence of structural variants in/near Maged1 or USP7, we focused on 175 nucleotide polymorphisms (SNPs). We investigated their potential associations with seven CUD-related phenotypes reflecting all stages of disease progression, correcting for multiple testing. The outpatient population comprised 77% men aged 38 +/- 9 years old. All the outpatients reported lifetime cocaine use and 72% of these men suffered from severe CUD (including to crack-cocaine). Comorbid SUDs were tobacco smoking (current for 89% of the sample), alcohol (62%), opioid (61%), cannabis (64%), and/or benzodiazepine (41%) use disorders. Half the sample had three lifetime SUDs or more (tobacco excluded).”

Second, we have added a description of the patients’ clinical profile as Extended Data Table 3.

6) It was not entirely clear if both males and females are being used. This is clearly an important issue with regard to sex as a biological variable, and cocaine demonstrating well known sex-specific effects in mice and humans. And Maged1 is an X-linked gene, so males and females should be examined.

We are sorry that this point was not made clearer for our Maged1 cKO mice. Thus, we modified our Materials and Methods section as follows: “Our experiments were conducted on hemizygous Maged1 KO males and their wild-type littermates (Maged1 WT) generated by crossing heterozygous Maged1 KO/WT females with C57Bl6J males as Maged1 KO males

are deficient in sexual behavior and do not reproduce properly (Dombret et al. 2012)” and “The Cre-mediated deletion of Maged1 was obtained using mice expressing Cre-recombinase under the control of the vGluT2 promoter, vGluT1 promoter and EMX1 promoter. As with the Maged1 KO mice, we used hemizygous Maged1 cKO males and their floxed littermates to maintain the same conditions allowing comparison of the KO and cKO types.”

Indeed:

- As described by Dombret et al. 2012 Maged1 KO males are deficient in sexual behaviour and do not reproduce properly (not due to morphological abnormalities, low sperm count or impairment of hypothalamo-pituitary-gonadal axis). To compare our results with those of our previous study (De Backer et al. 2018) and Maged1 cKO mice, we worked under the same conditions with hemizygous Maged1 cKO males.
- The presence of only one floxed Maged1 locus in the male avoids mosaicism due to the possible Cre recombination of only one floxed Maged1 locus in homozygous Maged1 floxed females (Vooijs, Jonkers, et Berns 2001; Song et Palmiter 2018).

Regarding differences in cocaine sensitivity between C57/Bl6J females and males, Eric Nestler's group (Calipari et al. 2017) showed that there was indeed a difference between the two sex but only when females are studied during Proestrus/estrus and not during dioestrus (I and II). The management of this parameter is very difficult during a sensitization protocol, which is another reason why we only used males. In the clinical sample, we showed that the associations between MAGED1 and transition from 1st cocaine use to CUD and between USP7 SNPs and cocaine-induced aggression, respectively, were not confounded by sex in Cox regression models (Extended Tables 5 and 6). Not only these associations remained significant when sex was included as a potential confounder in the model, but sex was not associated with the clinical outcomes per se. We also investigated associations between biological sex and both cocaine- induced aggression and transition from 1st cocaine use to CUD using bivariate analyses, showing neither was significant (cocaine-induced aggression mean =1.39 in women and 1.38 in men, median =0 and interquartile range =0-3 in both sexes, Mann-Whitney 4885, p-value = 0.8707; transition from 1st cocaine use to CUD median = 12, interquartile range =0-60 in women and 24 months, interquartile range =0-84 in men, log-rank test Chi2 =1.2, p =0.2). These data were added to the supplementary text.

Reviewer #2 (Remarks to the Author):

Cheron et al report a comprehensive analysis of the regulation of H2Aub in the mouse thalamus following cocaine treatment. The examination of histone ubiquitination in the context of cocaine exposure is a novel and valuable contribution to the field of neuroepigenetics. The manuscript is important in that characterization of H2Aub in this context is timely and highly rational, given that it is known to be regulated by cocaine in several brain reward regions. The manuscript includes a plethora of approaches to comprehensively interrogate H2Aub within the context of a putative H2Aub regulatory protein, Maged1, using a Maged1 conditional knock-out mouse line. These include cocaine locomotor sensitization behavior, RNA-seq, ChIPmentation, co-IP MS/MS, co-IP western blotting, and human genomics. The conclusions are extremely compelling and exciting, namely, (1) rigorous and innovative experimental approach for conditional knockout of Maged1; rationale for the PVT region of knockout is strong (2) comprehensive and robust analysis of the effect of Maged1 across multiple brain regions, including the robust attenuation of cocaine locomotor sensitization with glutamatergic neuron Maged1 cKO in PVT, (3) report an increase in H2Aub in several brain regions, (4) ChIP-sequencing of H2Aub in thalamus following cocaine treatment and (4) identification of human MAGED1 variants that associated with cocaine use disorder.

Enthusiasm for this manuscript is dampened by several major concerns, which are broadly listed below. Detailed feedback and requests for information and revisions are found in the accompanying annotated PDF.

We appreciate the reviewer's insightful comments and recognition of the novel contributions in our study. We have carefully considered the reviewer's feedback and have made necessary revisions to enhance the clarity and impact of our manuscript.

- It is unclear if the Maged1-cKO expresses a truncated Maged1. This is suggested by the the co-IP MS, for which the authors report a loss of only one co-IP'd protein between wild-type and Maged1-cKO condition. The expression of a truncated Maged1 protein in the cKO condition would require a major revision to the manuscript, with respect to (1) validation of the truncation and (2) conclusions drawn.

We thank the reviewer for this pertinent remark.

We tried to clarify this important point as soon as possible in the manuscript and we added some point of discussion regarding this matter (PL).

In the present study and in our previous paper studying Maged1 function (De Backer et al. 2018) we used Maged1-deficient (Maged1 KO) and conditioned (Maged1 loxP) mice that have been previously thoroughly generated, described and validated (Bertrand et al. 2008). These (c)KO mice are known to express a truncated Maged1 protein as shown on our blots. Our RNA sequencing results are also in line with this (Extended Data Fig 14). The hypothesis tested in this study were tested knowing that there was a truncated protein (deleting both MAGE Homology domains (MHD1 and 2 and other interacting Maged1 interacting domains) (Sasaki et al., 2005; Mouri et al., 2013)) still expressed in the (c)KO mice. We are now trying to make this less confusing earlier in our manuscript.

Indeed, we intentionally selected a commercial polyclonal antibody against Maged1 that immunoprecipitated Maged1 as well in both configurations, the normal Maged1 (control, untruncated) and the truncated Maged1 (inactivated or functional KO). See manuscript line 146, page 4: “we aimed at identifying Maged1 protein partners by performing coimmunoprecipitation and mass spectrometry (co-IP-MS) experiments with an anti-Maged1 antibody that recognizes both the full length and truncated form (functional KO) in saline- and cocaine-treated mice (Fig 2e).”

After IP-MS, we identified 404 potential Maged1-interacting proteins (see Supplementary Text and Extended Table 2). Our first hypothesis, to explain the major phenotypic difference at the end of a classical 5-day cocaine sensitization protocol, was that it might exist a protein partner that would differentially link to Maged1 especially under chronic cocaine administration but that would not link to the truncated form of Maged1. Following this hypothesis, we identified one protein, the histone H2A, that is a Maged1 partner after chronic cocaine administration and that does not link to the truncated form of Maged1 (following the same stringent statistical tests, confer methods).

When considering proteins that are linked to Maged1 in comparison with its truncated form, we also confirmed a well-known interaction between Maged1 and Praja1, Praja2 (Fig. 2f) (Sasaki et al., 2002). Remarkably, we observed that USP7 is a partner of Maged1 under both saline and cocaine conditions (Fig. 2f).

As mentioned above, the deleted region contains the conserved MAGE homology domain (MHD) as well as the Maged1 segment that is known to bind important partners such as p75NTR, Necdin, Ror2, Praja1/2, TrkA (Sasaki, et al. 2005). PCR confirmed that the Maged1 gene was successfully targeted and immunoblotting of brain lysates demonstrated that full-length Maged1 was absent in the deleted animals (full KO). When we used vGluT2^{cre}, Maged1^{lox^p} mice, we could still observe a residual amount of the normal Maged1 protein corresponding to its expected expression in the ~10% of non-vGluT2 (cfr. FACS results) cells in the thalamus. Exons 1–3 of the Maged1 gene remained intact in the deleted animals and we observed the presence of a truncated Maged1 fragment on immunoblots that presumably contained the initiator codon in exon 2 and the subsequent coding sequence in exons 2–3 (Fig. 2g-i). This portion of the Maged1 protein has no obvious protein motifs and is poorly conserved and analyses conducted by Bertrand et al. (2008) assumed that the mice lacking exons 4–12 of the Maged1 gene represent functional knockouts (Extended Data Fig. 14). However, while we acknowledge this limitation of our study as we are not studying a complete KO animal, we wanted here to discuss some reasons that reasonably exclude a dominant negative effect of the truncated protein in the phenotype and unraveled mechanism.

- 1) After re-expression of Maged1 in the PVT of KO mice we observed a strong and significant rescue of cocaine-induced locomotor sensitization (Fig. 1m). As the truncated protein is still expressed in the rescued mice, this result suggests a loss of function truncation rather than a dominant negative effect of the truncated protein. Indeed, a significant gain-of-function mutation would only be rescuable by a combination of WT gene augmentation with a knockdown of the truncated Maged1 protein (Zhao et al. 2021).
- 2) We observed that USP7, a PRC deubiquitinase, is a partner of Maged1 and that it is not a partner of the residual truncated Maged1 protein (Fig. 2f, right panel). We showed that USP7 inhibition (through P5091 administration) significantly alters cocaine-induced locomotor sensitization (administration of P5091), as well as the Maged1 cKO (Fig. g-j). This indicates that the mechanism sustaining the drug-related phenotype in KO mice involves a loss of function of the WT Maged1 (that is a partner of USP7) and not the truncated protein. Of course, it might still be possible that we are looking here at two different pathways (both controlling the same phenotype), one involving a truncated Maged1 protein and the other involving USP7 independently of WT Maged1. However, this latter possibility might seem far-fetched.

- Conclusions about the relative expression of Maged1 in PVT are vague and derive from unquantified, ‘qualitative’ imaging analyses.

Sorry about this oversight. We have now added the quantitative results in Figure 1i.

- The organization of the manuscript leads to some confusion (eg the abstract and results are not well-aligned; details of the truncation appear only in the second section although the cKO appears in the first section)

To increase clarity, we adjusted the abstract and discussed details on the truncation (please, see previous answers).

- The results sections do not include pertinent details that are necessary to interpret the findings (eg cocaine dose and treatment paradigm, sex and age of subjects)

Thank you for this remark, we are now trying to give more details in the result section. All our experiments were performed in 2- to 4-month-old mice. In the classical cocaine-induced locomotor sensitization: the three-first days of experiment, mice were injected with saline and their locomotor activity was recorded, the next 5 days, mice were injected with cocaine instead of saline (20 mg/kg, intraperitoneal (i.p.)). When specified, we performed a two-injection protocol (15 mg/kg, i.p.) with 3 days of habituation and 7 days between the two cocaine injections, based on Valjent et al. (2010).

- Female mice are excluded without a rationale provided

Indeed, we have not sufficiently justified our rationale for our choice and we apologize for this. As mentioned to Reviewer 1 the rationales are the following:

- As described by Dombret et al., 2012, Maged1 mutant males are deficient in sexual behaviour and do not reproduce properly (not due to morphological abnormalities, low sperm count or impairment of hypothalamo-pituitary-gonadal axis). To compare our results with those of our previous study (De Backer et al. 2018), we worked under the same conditions.

- The presence of only one floxed Maged1 locus in the male avoids mosaicism due to the possible cre recombination of only one floxed Maged1 locus in homozygous Maged1 floxed females (Vooijs, Jonkers, et Berns 2001; Song et Palmiter 2018).

We thus modified our Materials and methods section as follows: “Our experiments were conducted on hemizygous Maged1 KO males and their wild-type littermates (Maged1 WT) generated by crossing heterozygous Maged1 KO/WT females with C57Bl6J males as Maged1 KO males are deficient in sexual behavior and do not reproduce properly⁵⁰.” and “The Cre-mediated deletion of Maged1 was obtained using mice expressing Cre-recombinase under the control of the vGluT2 promoter, vGluT1 promoter and EMX1 promoter. As with the Maged1 KO mice, we used hemizygous Maged1 cKO males and their floxed littermates to maintain the same conditions allowing comparison of the KO and cKO types.”

An additional expert on human genomics must necessarily review the CUD association study methods and results. Extremely powerful component of this manuscript.

In response to this request, we asked Gilbert Vassart and Michel Georges to proofread our manuscript. They are both experts in this field, as their track record attests. We have duly acknowledged their names in our acknowledgments section.

Comment list and answers from pdf document “Reviewer 2 marked up file.pdf”

- 1) Recommend revising the abstract to reflect the data as presented in the manuscript. That is, first maged1-KO transcriptomics, then MS, then examination of H2Aub, based on the results of those experiments.

We intentionally decided to write this abstract to emphasize on the main biological discoveries of this study rather than describing step by step in the exact same order the experiments that conducted to the results. Still, we do agree that this comment raises an important point of confusion if the paper doesn't look like to correspond to the abstract. We thus tried to make it clear that the epigenetic (and central) discovery was made while we were investigating the mechanisms of our initial gene of interest, Maged1.

- 2) The examination of histone ubiquitination in the context of cocaine exposure is a novel and valuable contribution to the field of neuroepigenetics.

- 3) Vague. Does this mean at genes localized to the H2Aub? Or local to the PVT?

Sorry for this lack of specificity. We meant local to the thalamus. To avoid confusion and unnecessary complexification in the abstract, we decide to remove that term.

- 4) Specific relative to what? Brain region, hPTM, cocaine? Perhaps remove this qualifier in the abstract.

We agree and we removed it. Thank you.

- 5) abolished or attenuated?

The sensitization per se is abolished (please see R1 response, Fig. 3g, Extended Data Fig. 10a (not significant Sidak's posttest (D1 versus D7) for vGluT2cre Maged1loxp, and $**P = 0.0077$ for control mice), and our previous paper, De Backer, et al, 2018).

- 6) Extremely valuable and compelling.

- 7) Revise to

1. expand upon the rationale for the selection of Cre-driver lines

2. describe the anatomical localization of the maged1 ko; including telecephalic neurons and circuit between NAc, Amy, VS, PVT

In our first paper on Maged1 we showed that Maged1 is not necessary in GABAergic cells,-and dopaminergic cells for drug-addiction related behaviors (De Backer et al, 2018). We then demonstrated that it was partially necessary in glutamatergic nuclei

like the PFC and the amygdala. Here we confirmed that expression of Maged1 in glutamatergic and specifically vGluT2 cells was necessary and sufficient for drug-adaptive behaviors.

8) Rigorous and innovative experimental approach for conditional knockout of Maged1. Rationale for the region of knockout is strong.

9) 2. Co-IP experiments report that this cKO contains a truncated Maged1. Revise to

1. report the details of the truncation in this section,

2. report the expected function of the truncated Maged1 (ie what functional domains are retained? what number of exons?)

Please, refer to our detailed answer above.

10) Fig 1f: Does this statistic validate that there is no effect of day on locomotor behavior in the KO? It appears that the KO does sensitize, but authors state that sensitization was 'abolished.' Please confirm statistically or revise to 'attenuated.'

Please, refer to our detailed answer above.

11) Fig 1c, f include 'maged1' KO data. Revise results and figure legends to define this data.

Thank you, we appreciate your keen observation. We have now described this data in the legend.

12) Revise to report the results of these experiments.

This sentence was added to the manuscript: "Indeed, specific inactivation of Maged1 in telencephalic Emx1 cells did not alter cocaine-induced locomotor sensitization (Extended Data Fig. 2)."

13) Comprehensive and robust analysis of the effect of Maged1 across multiple brain regions.

14) How much of the thalamus is covered by the PVT? Is this analysis specific enough relative to the maged IHC that was specific to the PVT?

Considering the limited size and number of cells of the PVT to perform RNAseq (and CHIPmentation), whole thalamus samples were preferred over PVT dissection to limit the number of mice needed (respecting the 3Rs rules, mandatory for Ethical Committees in Europe), and speed-up sample processing, minimizing RNA degradation (and alteration of H2Aub dynamic).

15) Revise to

1. report the number of genes in the maged1-cko (as is reported above for the control mice).
2. report the percent overlap of the saline vs cocaine DEG and FET
3. report the total number of transcripts measured in each condition in each line

This information can be found in the supplementary text and Extended Data Table 1.

16) What is meant by 'curated gene set'?

Gene sets in this collection are curated from various sources, including online pathway databases and the biomedical literature. They can be found here: <https://www.gsea-msigdb.org/gsea/msigdb/human/genesets.jsp?collection=C2>.

Of course, when we found a significant overlap with a set of gene, its source was referenced.

17) a) Unclear how the authors draw this conclusion.

Actually, we did not find any significant overlap between the genes that were similarly regulated (significantly up- or down-regulated upon cocaine) in the control and in the cKO with genes sets related to PRC2. We have now added a clearer sentence.

b) Unclear how this relates to the data. What is this an example of? What mechanism is being speculated upon? Revise for clarity, and consider moving to the discussion

Indeed, we modified and displaced this sentence as its purpose is to make things clearer.

18) Unclear from where the data are derived. Are the hPTM enrichments from ENCODE? Do they match species/age/sex/tissue of the Maged1-cKO and control tissue used for RNA-seq? If not, how do the authors interpret the meaning of the comparisons?

The data were obtained from the GSEA software, a joint project of UC San Diego and Broad Institute (Subramanian et al. 2005). All the tested gene sets, along with their sources references and statistics, can be found in table 1 (under the curated gene sets tab).

19) Did the authors validate the specificity of the antibody using the Maged1 KO mice?

Yes, we did (Fig. 2e).

20) What brain region was used for the co-IP? Also PVT?

Like for RNAseq and ChIPmentation, we used the whole thalamus for the same reasons mentioned above.

- 21) A very interesting finding. The authors have identified histone modifying enzymes (PRC complex) associated with Maged1 in brain.
- 22) Confusing and potentially problematic for the conclusions drawn. Presumably all of the Maged1 interactions would be disrupted in the Maged1 cKO samples? Why or why not? Do the authors suggest that most Maged1 interacting proteins interact with the truncated Maged1? If so, the cKO condition must be revised to accurately present the findings as a condition of a truncated protein rather than a KO condition. Truncated Maged1 may act as a dominant negative, and at the very least may have a phenotype that is different than a 'true' cKO condition.

Please, refer to our detailed answer on the truncated protein above.

- 23) Cocaine was administered to mice before thalamic mononucleosome preparations? What was the treatment paradigm?

Indeed, cocaine was administered to mice in vivo. The treatment paradigm can be found on Fig 2a and 3a and also in the Methods section under protein extraction paragraph.

- 24) Also from mononucleosome samples or from whole nuclear protein extract? Are mononucleosomes crosslinked prior to co-IP?

Yes, from mononucleosomes that were not crosslinked prior to co-IP.

- 25) Unclear how this conclusion is drawn based on the images in Ext Fig 7. There is fluorescence throughout the section (especially DG and CA1 regions). No quantification of fluorescence intensity is performed. Recommend revising 'higher in the PVT' to 'expressed in the PVT' as well as other brain regions.

Sorry about this oversight. We have now added the quantitative results in Figure 1i.

- 26) Revise to explain rationale for this hypothesis. Is USP7 an E2/3 ubiquitin ligase? How many proteins in addition to Mage1 assemble with USP7? Why is Mage1 prioritized? Does Mage1 co-IP with Maged1? If not, why not?

USP7 is a PRC member and a deubiquitinase. Mage proteins are known to be partner of different E2/3 ubiquitin ligases and thus play an important role in different ubiquitination pathways. PRC complexes are formed by different E2/3 ubiquitin ligases and deubiquitinases like USP7. Finding a link between USP7 and Maged1 suggest that Maged1 might be needed for H2A (de)ubiquitination.

- 27) How was this antibody validated for specific IP of H2Aub?

The H2Aub antibody used in this study has been used and validated previously (e.g. <https://doi.org/10.1371/journal.pgen.1000506>). The genome-wide distribution of H2Aub enriched regions closely parallels that of other studies (see Extended Data Fig. 16 in which reanalyzed available H2Aub coverages from Kallin et al. 2009) with a high enrichment at gene promoters. Besides, this distribution differs from the ubiquitous and promoter enriched distribution described for H2A (e.g. Zhang, Roberts, et Cairns 2005) supporting that our H2Aub antibody specifically recognize this epigenetic modification rather than non-specifically immunoprecipitating H2A-bound chromatin.

- 28) Is there any data quantifying H2Aub in PVT following cocaine treatment? If not, This conclusion can not be drawn from the data. The interaction of these proteins does not suggest that they function in the cocaine condition any differently than in the saline condition.

Here we wanted to guide the reader through our reasoning and not to over-interpret data. We thank the reviewer for this remark. Thus, we rephrased this sentence by: 'From these newly demonstrated interactions we hypothesized that'

- 29) Fig 3 reports Maged1 KO (not cKO) for the NAc data. Please revise for accuracy/clarity.

Yes, to ensure a complete inactivation of Maged1 in the NAc we used Maged1 full KO mice.

- 30) Very valuable finding regarding the regulation of H2Aub in brain reward regions following cocaine exposure.

1. Can the authors confirm that this result is novel?
2. What is the treatment paradigm?

Yes, we confirm that to the best of our knowledge this result is novel. The treatment paradigm consisted of 5 days of cocaine administration before proteins extraction (Fig. 3a).

3. Why is ChIPmentation preferred over conventional ChIP-seq or CUT&RUN? How does it differ from conventional ChIP?
4. Why is thalamus analyzed rather than PVT?

We acknowledge the pertinent remarks of the reviewer regarding the ChIPmentation experiments and answer to the concerns raised below:

- Indeed, this is the first report- to our knowledge- analyzing H2Aub dynamics in the thalamus or other brain regions related to cocaine or drug exposure. Besides, while other works addressed changes in the related H3K27me3 epigenetic mark genome wide analysis have not been previously reported (Cheron & de Kerchove d'Exaerde,

2021). Here we decided to focus in H2Aub coverages since H2A was identified as the differential Maged1 interactor and because of the reported role of Maged1 as regulator of the PRC1 activity (the complex in charge of H2Aub deposition).

- ChIPmentation is an implementation of the conventional ChIPseq methodology which improves the sensitivity of the method (Schmidl et al. 2015). This is achieved by introducing nextera adapters (required for the next-generation-sequencing of the sample) in the immunoprecipitated DNA directly after immunoprecipitation and washing of the chromatin/ antibody/ beads complexes using the Tn5 Tagmentase enzyme. In the conventional ChIPseq, the DNA is decrosslinked and purified after chromatin complexes immunoprecipitation and washes and subsequently used for library generation by adapter ligation. This involves the repair of the DNA ends, addition of A protruding ends and adapter ligation by the T4 DNA ligase and purification prior to PCR mediated library generation. Thus, the high efficiency of the Tn5 transposase compared to the conventional adapter ligation combined with the elimination of the described DNA end repair/ adenilation and adapter ligation/ purification steps allows to perform ChIP experiments from relatively small samples without affecting the sensitivity of the technique. This aspect was very important for us since it allowed to considerably reduce the number of animals required for the study, since our experimental design involved four experimental conditions to be strictly processed in parallel. The Cut and Run methodology did not work well in our hands. Besides, since H2A K1119ub is a transient epigenetic mark deposited during the polycomb-mediated epigenetic repression, we were concerned that the requirement of performing CUT&RUN in dissociated living cells could affect H2Aub coverages in our samples.

- The H2Aub antibody used in this study has been used in ChIPseq experiments in a number of previous works (e.g Kallin et al. 2009). The genome- wide distribution of H2Aub enriched regions closely parallels that of other studies (Kallin et al. 2009; Yang et al. 2014) with a high enrichment at gene promoters (Extended Data Fig. 16). Besides, this distribution differs from the ubiquitous and promoter enriched distribution described for H2A (e.g, (Zhang, Roberts, et Cairns 2005), supporting that our H2Aub antibody specifically recognize this epigenetic modification rather than unspecifically immunoprecipitating H2A-bound chromatin. As shown in Extended Data Fig. 16, H2A is broadly distributed genome-wide with only mild enrichment over gene bodies and is depleted from the nucleosome free regions of gene transcription start and transcription termination sites. Instead, H2Aub is mostly enriched at gene promoters. Thus, the differences in our H2Aub coverage profiles with those reported for H2A and their resemblance to the H2A distribution of Yang et al strongly suggests that the H2Aub enrichments reported in our study are specific. This discussion has been added in the supplementary text.

Whole thalamus samples were preferred over PVT dissection to facilitate and speed-up sample processing since, as mentioned, above we were concerned that rapid H2Aub dynamics could affect the interpretation of the results. Although this may partially mask the differences between our experimental conditions, we believe it does not globally affect our interpretation of H2Aub dynamics across them.

31) Unclear to what this refers in this section. There does not appear to be a Maged1 inhibitor on board in any experiments.

Here, we are talking about the functional KO that inactivate Maged1. This term is clearly defined in the paragraph below.

- 32) To validate their hypothesis that Maged1 functions via its interaction with USP7 to regulate H2Aub in the cocaine condition, the authors should confirm that there is no effect of USP7 on cocaine sensitization in the Maged1 cKO condition. Should the inhibitor block sensitization in the cKO condition, then authors can conclude that it functions downstream or in parallel to Maged1. Revise to report the results of such an experiment and/or discuss as a rationale and future direction.

We agree with the reviewer that to further exclude any possible effect of the USP7 inhibitor downstream or in parallel to Maged1 we had to exclude an effect of P5091 on sensitization of Maged1 cKO mice. We did not observe any difference in locomotor activity during the two cocaine injections protocol between the cKO administered with vehicle and the cKO that were injected with P5091 (Extended Data Fig. 10c).

- 33) Why do the authors use a shorter sensitization paradigm for these experiments, and a longer paradigm for the cKO experiments?

For these experiments, involving a preclinical drug injection (P5091), we decided to slightly modify the classical sensitization protocol and we used the two-injection protocol (described in this paper: Valjent, et al., 2010). There are two main reasons for that: (1) the sensitization effect is demonstrated to be higher after a week interruption between the 2 injections of cocaine – potentially increasing the sensitivity for an effect of a preclinical drug (Valjent, et al., 2010) and (2) it allowed a longer time period between the 2 cocaine injections to deliver a preclinical drug lowering the risk of pharmacological interaction. In this two-injections protocol we recorded the locomotor activity for 30 minutes (Fig. 3) instead of the classical 60 minutes (Fig. 1).

- 34) Please confirm that (1) the Maged1-cKO data shown in figure 3 are a different cohort than that of figure 1 (2) the Maged1-cKO and P5091 experiments were run simultaneously.

- (1) We confirm that these experiments were not done with the same animals (Fig. 1 vs 3). These different cohorts of mice (data shown on Fig. 1 and 3) went through different experimental paradigms.
- (2) We confirm that we conducted experiment on vGluT2cre, Maged1 loxp and P5091 using the protocol under the same conditions.

- 35) continued what? the P5091 injections?

Continued the same experiment with P5091 and cocaine injections. We consequently slightly modified the text.

- 36) 1k: what is the fluorescence in the dentate gyrus and why is it not from the retrograde AAV?

This is from nuclear staining (hoechst).

37) Fig 2: resolution too low, many labels are illegible

This issue should be solved with higher resolution figures that will be uploaded.

38) Label the maged1 band with an arrow in the MWM. What is the band at ~35 kd that appears only in the KO? Is that the truncated Maged 1?

Thank you, Maged1 band is now indicated with a gray arrow and Maged1-KO truncated protein is indicated with a blue arrow on Fig. 2c

39) Why is there a band at the same size as MAGED1 in the KO condition?

The answer to this question can be found in detail in the Methods section. To generate Maged1 KO cell lines we used the RNA-guided CRISPR-Cas nuclease system (Ran et al. 2013). By analyzing the PCR result the combo 6 (gRNA 2+6) was chosen as the best combination to delete the region between exon 2 and 5, even if a wild type band was still present. This was expected since HEK293T cells have 3 X chromosomes containing the MAGED1 gene, so the heterozygous clone selected (clone A1) was re-transfected with Cas9 plasmid + gRNA 2+6, two isolated sub-colonies were obtained and the abolished expression of MAGED1 at protein level was confirmed by WB (Fig. 2k). Finally, and because of slight Maged1 residual expression in B9 clone, the sub-clone C1 was selected and used for subsequent experiments.

40) Unable to find the methods for western blot quantification in Fig 3. Revise to include quantification and statistics for western blots in this section; or add a heading to the other sections.

We are sorry about this forgetfulness. We have added the appropriate and missing section in the Methods section, western blot paragraph.

Reviewer #3 (Remarks to the Author):

Review for Nat.Comm. 2023: Histone H2A monoubiquitination in the thalamus regulates cocaine effects and addiction risk

The end goal of this study is to provide new risk factors and therapeutic targets for cocaine use disorder (CUD). In a previous study the authors identified Maged1 as a master regulator of cocaine reward and reinforcement in mice. Therefore, they centered this work on the characterization of the molecular mechanisms involving Maged1 in these cocaine-related phenotypes. The authors claim that locomotor sensitization in response to chronic cocaine usage requires the expression of Maged1, specifically in vGluT2 neurons and particularly in the paraventricular thalamus (PVT). Furthermore, Maged1 inactivation mainly led to an altered transcriptional regulation in the thalamus after cocaine administration. Through IP-MS experiments two of Maged1 partners were validated, namely: USP7 and H2A, the latter exclusively upon cocaine administration. Using human cell lines, the authors further confirmed the conserved nature of Maged1 and USP7 binding. Importantly, USP7 inhibitor recapitulated the loss of locomotor sensitization and was accompanied by an increase in H2Aub baseline levels. Interestingly, H2Aub thalamic ChIP-sequencing showed that cocaine administration in control mice increased H2Aub peaks and that genes displaying H2Aub cocaine-enrichment did not show enrichment in Maged1-cKO animals, stressing the link between cocaine dependent H2Aub gene regulation and Maged1 mechanism of action. This manuscript describes the different roles of Maged1 in different regions in the context of cocaine usage and for the first time its function in the PVT neurons has been addressed. The authors provide new sequencing datasets that support the hypothesized link between transcriptional repression observed as a consequence of cocaine administration, and H2Aub epigenetic control of such transcriptomic effect. While this work correlates H2Aub genomic regulation to CUD, it does not causally address its role in the development or maintenance of its behavioral feature of locomotor sensitization. Thus, already the title is misleading. The authors most of the time tend to overstate their conclusions and relate them to the epigenetic regulation, however most of the times their findings mostly focus on Maged1 and USP7 mechanisms of action. We advise for a cautious rephrasing of the key statements of this manuscript and the following major and minor revisions.

We greatly appreciate the reviewer's comprehensive evaluation and constructive feedback on our study. Our primary objective is to identify new risk factors and therapeutic targets for cocaine use disorder (CUD), focusing on the role of Maged1 as a central regulator of cocaine reward and reinforcement and its involvement in a previously undescribed epigenetic PTM in the field of drug addiction. We acknowledge the reviewer's observation regarding potential overstatements, and we will diligently rephrase key statements and title to ensure accuracy. We will carefully address the reviewer's suggestions and incorporate the advised major and minor revisions to enhance the precision and clarity of our manuscript.

Major

Fig. 1

In the text it is written that (73-77 lines) “To screen for potential vGluT2 nuclei involved in acquisition of this phenotype, we performed AAV retro mCherry injection in the NAc and revealed the amygdala, the ventral subiculum and the paraventricular thalamus (PVT) as the main vGluT2 nuclei targeting the NAc (Fig. 1g,h). Of these, Maged1 mRNA in situ hybridization and immunohistochemistry (IHC) showed a singularly high level of Maged1 expression (Fig. 1i) in the PVT, which has recently been shown to be involved in drug addiction.” From the results reported, we only see a zoomed in image of PVT Maged1 expression, but there are no quantifications validating the claim of higher expression of this protein in the PVT compared to the other regions. Additionally, from Extended Data Fig.1 it seems that Maged1 is ubiquitously expressed across the brain rather than a singularly high concentration in the PVT nucleus. Either rephrase or provide quantifications.

Thank you for this remark. The images shown in Extended Data Fig.1 are not perfectly medial and thus does not show specifically the PVT. To fix this oversight we have now added the quantitative results in Figure 1i.

Extended Data Fig.4/5

Panel d shows that the abolishment of locomotor sensitization found in the full Maged1 KO mice model was recapitulated by either the conditional-KO of Maged1 in vGluT2 or PVT neurons, but not other brain nuclei (see Extended Data Fig.4, panel d). This finding is partially in contrast with their previous discovery where the cKO of Maged1 in the mPFC led to an impaired phenotype using the same behavioral protocol (De Backer JF et al., EMBO Rep, 2018). The difference in these results has not been addressed in this study. Moreover, from the PFC and AMG re-expression experiment reported in Extended Data Fig. 5 (panel a and b) it seems that Maged1 rescue in a full KO model partially restores the cocaine sensitization behavior. To conclude that the PVT is the key nucleus necessary and sufficient for this phenotype a direct comparison to the Fig.1 panel m dataset would be required.

We thank the reviewer for this pertinent remark. Indeed, in our previous paper (De Backer, et al., 2018), we have shown a partial and significant effect after Maged1 inactivation in the amygdala and PFC. Still, this effect was very partial. Here we tried to represent in an integrated way all the results that we obtained from these different nuclei (PFC, ventral subiculum, amygdala, PVT) (Extended Data Fig 4). Even though statistical significance could be demonstrated when looking at these nuclei separately and in details (as shown in our previous paper), looking at them altogether (looking at a 5-days mean, Extended Data Fig. 4d) highlighted a stronger and significant effect for the PVT and vGluT2 neurons. In addition, in Fig. 1 panel m we showed that there is no statistical difference with the control while there is a statistical difference between control group and the rescue in the amygdala and in the PFC (Extended Data Fig. 5). We did not conclude that the PVT is the only necessary and sufficient nucleus. However, from our data, we are convinced that the PVT is necessary and sufficient for this sensitization phenotype and that it detached clearly from all the others vGluT2 and vGluT1, EMX1 nuclei (Extended Data Fig. 4d).

While revising this manuscript, to further test the role of Maged1 within vGluT2 neurons in other behavioral paradigms we conducted a CPP (Extended Fig. 3a) that is more suited to test rewarding properties of drug of abuse. In this experiment, we observed that vGluT2-mediated Maged1 inactivation led to a significant reduction in the time spent in the cocaine-paired compartment (Extended Data Fig. 3a). It should be noted that the various inactivation (PFC, Amygdala) in our previously published study (De Backer et al. 2018)⁹ did not show any effect on the acquisition of CPP. This contrasts starkly with the present finding involving Maged1 inactivation in vGluT2 neurons (Extended Data Fig. 3a), which emphasizes the clear influence of Maged1 in vGluT2 neuron on this behavioral paradigm.

Fig. 2

Adjust Fig.2 panel a to more clearly described the mice line used in this experiment. There is no information regarding the mouse line used to evaluate the efficiency of the FACS sorting procedure. Moreover, since for the RNA-sequencing experiment no FACS sorting was used, it is unclear which mouse line was processed. Additionally, is there any marker used to assess that the manual dissociation of the thalamus was precisely leading to a clean PVT isolation?

Sorry about this mistake, we mixed FACS experiment with the actual RNA-sequencing experiment. Now you can find the information for the mice line used in this experiment in Fig. 2 panel a and the information regarding FACS in Extended Data Fig. 6. Whole thalamus samples were preferred over PVT dissection to facilitate and speed-up sample processing

since we were concerned about RNA degradations and H2Aub dynamics alterations that could affect the interpretation of the results (see also answers 14 and 30 to Reviewer 2 marked up file.pdf)

Panel c-d related: These graphical representations do not back up the authors claim “Interestingly, a gene set enrichment analysis with curated gene sets revealed that cocaine-regulated genes in control but not in Maged1-cKO mice were targets of the Polycomb repressive complex (PRC) (Fig. 2c,d and Extended Data Table 1). Notably, the genes with expression similarly regulated by cocaine in the control and Maged1-cKO mice were not targeted by PRC action.” While from the curated gene sets plot it is possible to deduce that around one third of the downregulated genes after cocaine treatment are part of the PRC2 targets in control conditions, the same information is not reported for Maged1 cKO mice.

Following the alluvial plot, all the genes in between the dashed lines (the one that were analyzed in the curated gene sets) are genes that are (up or down-) regulated after cocaine administration only in the control and not in the cKO. These genes are known to be targeted by the PRC. The genes below and above these ones do not show any overlap with known PRC target genes.

Additionally, the bottom illustration should include information concerning the expression levels related to each of the proteins depicted and how this value changes across experimental conditions; thus better highlighting the involvement in Maged1 with PCR key players.

This is an illustration based on the literature (Schuettengruber, et al., 2017, Margueron et al., 2008, de Napoles et al., 2004). Its goal is to illustrate what are the PRC members and what they are known to do, as it seems that the genes we are looking at are known targets of the PRC.

The change in GO hits reported in Extended Data Fig.7 is not enough to describe which pathways are the most affected in the cKO model. Similar to Ext. Data Fig.5 panel d, a GO for cKO mice should be included.

We agree with the reviewer that we did not illustrate in a panel the most relevant pathways that are changing upon cocaine in the cKO. This detailed information can be found in the Extended Table 1. For down-regulated genes in cKO mice, we did not find any of the illustrated pathways for the control mice (Extended Data Fig. 6). Conversely, we could observe, at different rank and among other general GO terms, all the GO term illustrated for upregulated genes in the control mice. This result is in line with the idea (further illustrated in Extended Data Fig. 7) that Maged1 exerts its specific effects on the behavioral response to cocaine primarily through drug-induced downregulation.

Panel g-i related: To test whether Maged1 interaction with H2A and USP7 contributes to the regulation of the monoubiquitylation of H2A at lysine 119 (H2Aub) the authors use IP experiments and conclude that “these data suggest that Maged1 plays a role in H2A monoubiquitination in thalamic neurons following cocaine intake, via Maged1 involvement with a PRC1 member”. This interpretation is an overstatement as what the experiment shows is that the binding between USP7 and H2Aub depends on the presence of Maged1, however, per se the H2Aub PTM is present with or without Maged1 as confirmed in Fig.2h. Importantly, as Fig.2h and 1 refer to saline conditions, no conclusion can be drawn about cocaine effect on such PTMs, but only confirm that the binding between H2Aub, USP7 and Maged1 is not altered by cocaine consumption (see Extended Data Fig. 9).

Indeed, we corrected this overstatement: “From these newly demonstrated interactions we hypothesized that Maged1 plays a role in H2A monoubiquitination in thalamic neurons following cocaine intake, via Maged1 involvement with a PRC1 member.” Thank you for this pertinent remark.

Furthermore, it is not clear what the starting material for the Co-IP experiment was.

Sorry about this forgetfulness. We added a line specifying that the thalamus was extracted for Co-IP where it was lacking: “we aimed at identifying Maged1 protein partners in the thalamus by performing coimmunoprecipitation and mass spectrometry (co-IP-MS)” This was already specified below in the text: “To confirm these interactions, we performed Maged1 co-IP with mononucleosome samples extracted from the thalamus, followed by...” and this information can also be found on panel 2a referring to panel e-i.

Extended Data Fig. 8

To reinforce the author’s claim “we observed that Maged1 and USP7 protein levels were both higher in the PVT than in other parts of the thalamus”, a thorough quantification of Maged1 and USP7 in all thalamic nuclei should be included.

We added this analysis, using 5 independent mice (Extended Data Fig. 8). We have shown that Maged1 and USP7 protein levels were both higher in the PVT than in the adjacent mediodorsal thalamic nucleus (MDT) (including its medial, lateral and central parts) and the intermediodorsal nucleus of thalamus (IMD). We are therefore refining our sentence to provide greater accuracy, referring to that specific quantification rather than a more subjective qualitative claim.

Fig. 3

Panel b-e: from the representative WB view reported in panel b, it seems that H2A has a high variability across conditions and therefore cannot be used as a normalizer in the following analysis (c-e). Compare the results obtained with a B-actin normalization and/or an additional protein known not to be subjective to Maged1 and cocaine manipulations. Please include (as supplementary) the representative WB images relative to NAc and PFC as well. Could an IHC for H2Aub specifically directed at the vGlut2 PVT neuronal population be added in support of this findings?

We apologize for this oversight. The representative images for the KO group appeared dimmer probably due to the JPEG compression. As more evident from our raw western blot data (Extended Data Fig. 15) and new representative images in Fig. 3, we didn't observe significant variability when comparing the absolute levels of H2A/B-actin. Still, it is because there might be a variability in histone extraction that H2A is the most appropriate normalizer to H2Aub. We thought that using B-actin or another classical normalizer that is not sensitive to DNA degradation or chromatin-linked protein extraction was not appropriate. This normalization method through H2Aub/H2Atotal ratio has been previously validated as a way of controlling WB PTM bands intensity (Bravo, et al., 2015, Zhang, at al., 2017) and is similarly used for analyzing other classic PTM (Buttress, et al, 2022). Moreover, we could recapitulate the thalamic increase (or not in the cKO group) in H2A monoubiquitination found in our WB analysis in the ChIP-seq experiment using the only H2Aub antibody (Fig 3i-q). Thus, this reasonably excludes that the changes in mono-ubiquitination we observed would be artifacts of the WB normalization process. Regarding your IHC for H2Aub inquiry, we must admit that in our hand, for IHC, the H2Aub antibody was not sensitive enough to find any IF differences between conditions.

Line 177

“These results showed that pharmacological abolition of cocaine-evoked H2Aub deposition in the thalamus impeded cocaine-evoked behavioral sensitization.” This conclusion should be reconsidered as it stems from the USP7 pharmacological inhibition. Therefore, by blocking USP7 deubiquitinase activity, the sensitized behavior is altered. The result not mentioned but worth discussing, is that saline treated mice have a higher H2Aub levels. Could it be possible that a further increase due to cocaine administration is contrasted due to ceiling effects? Additionally, what could be the biological impact of such a higher baseline in H2Aub levels? In essence, any strong implication of H2Aub deposition and their causal role in the cocaine evoked phenotype is an overstatement as this could be only hypothesized due to the correlation between the two phenomena. Finally, how specific is the USP7 inhibitor? There is no evidence for its specificity and important controls are missing.

In line with your concerns about overstatement we decided to write this sentence with a more balanced and neutral tone, focusing on the observed relationship without overstating the causal effect: “These results showed that pharmacological abolition of cocaine-evoked H2Aub deposition in the thalamus paralleled an alteration of cocaine-evoked behavioral sensitization”.

Indeed, we agree that this ceiling effect would be a possible explanation and “graphically” this is also what we first thought. Although it could not be demonstrated statistically. The interaction factor was the only one to exhibit a Pvalue below 0.5 (please see legend of Fig. 3 panel e), and even looking at a Sidak’s multiple comparisons test did not demonstrated significance. Indeed, as you can see on this table referring to panel e, the difference between the saline level of H2Aub was not statistically different.

	Predicted (LS) mean diff,	95,00% CI of diff,	Below threshold?	Summary	Adjusted P Value
Šídák's multiple comparisons test					
Saline:Ctrl vs. Saline:Vg2 cre Md1 flox	-0,2234	-0,5331 to 0,08638	No	ns	0,2854
Saline:Ctrl vs. Cocaine:Ctrl	-0,3022	-0,5824 to -0,02195	Yes	*	0,0281
Saline:Ctrl vs. Cocaine:Vg2 cre Md1 flox	-0,1821	-0,4687 to 0,1044	No	ns	0,4284
Saline:Vg2 cre Md1 flox vs. Cocaine:Ctrl	-0,07882	-0,3653 to 0,2077	No	ns	0,9744
Saline:Vg2 cre Md1 flox vs. Cocaine:Vg2 cre Md1 flox	0,04122	-0,2515 to 0,3339	No	ns	0,9993
Cocaine:Ctrl vs. Cocaine:Vg2 cre Md1 flox	0,12	-0,1412 to 0,3813	No	ns	0,7684

Regarding USP7 inhibitor specificity, it is possible that it exerts its effect on cocaine sensitization through a parallel pathway (than the one highlighted here). Thus, we administered P5091 to cKO mice and showed that there was no additive behavioral effect on cocaine sensitization (Extended Data Fig. 10c). This control experiment reasonably excludes an off-target effect of USP7 inhibitor on locomotor activity.

Human data:

Has the patient cohort been controlled for other types of medication?

We have also collected the number of both psychotropic and non-psychotropic medication used by the patients at the time of interview. We did not include it in the model testing cocaine-related phenotypes, since the latter were considered on a lifetime basis. Yet, we acknowledge this remains a relevant information for the reader for the sake of sample description. We have added this information in a new table, Extended Data Table 3. Moreover, no measure of medication regimen was significantly associated with the transition from 1st cocaine use to CUD nor cocaine-induced aggression (data not shown).

Minor:

Extended Data Fig.4: report group size for experiment panel d.

It is thoroughly reported.

Fig.2: panel e, add on top of each column the IP antibody to help the reader navigate the WB. Thank you for this remark that is in accordance with reviewer 2. we now have added arrows to help the reader read the WB.

Fig.2: panel f, the order of the volcano plots and the corresponding legend descriptions are inverted. The legend used on the upper part of the volcano plots are misleading, please correct. (the left and right ends of the plots do not correspond to the groups highlighted with the labels but they reflect a higher or lower expression compared to the reference group used). Sorry for this misleading legend, right and left were inverted. We have corrected this mistake and removed the misleading legend on the top.

Fig.2: panel l-m same as before, legend used on the upper part of the volcano plots are misleading. We have corrected the misleading legend.

Line 144 add Fig number. Thank you for spotting this detail. It has been corrected.

Fig.3 panel K: What has been used to normalize H2Aub values? Any representative images? β actine and H2A have been used to normalize H2Aub values, as shown on representative images Fig. 3b.

Fig.3 panel O: wrong control labelling. Indeed, thank you for spotting that detailed mistake. It is now corrected.

Line 246, discussion section, the word “complete” is inappropriate especially when referring to the restoration experiment where only a partial rescue of cocaine sensitization was obtained.

Indeed, even though there is no statistical significance demonstrated here it does not mean that the restoration is complete. We thus removed this word that might be overstating the situation.

References

- Bertrand, M. J. M., R. S. Kenchappa, D. Andrieu, M. Leclercq-Smekens, H. N. T. Nguyen, B. D. Carter, F. Muscatelli, P. A. Barker, et O. De Backer. 2008. « NRAGE, a P75NTR Adaptor Protein, Is Required for Developmental Apoptosis in Vivo ». *Cell Death and Differentiation* 15 (12): 1921-29. <https://doi.org/10.1038/cdd.2008.127>.
- Calipari, Erin S., Barbara Juarez, Carole Morel, Deena M. Walker, Michael E. Cahill, Efrain Ribeiro, Ciorana Roman-Ortiz, et al. 2017. « Dopaminergic Dynamics Underlying Sex-Specific Cocaine Reward ». *Nature Communications* 8 (1): 13877. <https://doi.org/10.1038/ncomms13877>.
- De Backer, Jean-François, Stéphanie Monlezun, Bérangère Detraux, Adeline Gazan, Laura Vanopdenbosch, Julian Cheron, Giuseppe Cannazza, et al. 2018. « Deletion of Maged1 in Mice Abolishes Locomotor and Reinforcing Effects of Cocaine ». *EMBO Reports* 19 (9). <https://doi.org/10.15252/embr.201745089>.
- Dombret, Carlos, Tuan Nguyen, Olivier Schakman, Jacques L. Michaud, Hélène Hardin-Pouzet, Mathieu J. M. Bertrand, et Olivier De Backer. 2012. « Loss of Maged1 Results in Obesity, Deficits of Social Interactions, Impaired Sexual Behavior and Severe Alteration of Mature Oxytocin Production in the Hypothalamus ». *Human Molecular Genetics* 21 (21): 4703-17. <https://doi.org/10.1093/hmg/dds310>.
- Kallin, Eric M., Ru Cao, Raja Jothi, Kai Xia, Kairong Cui, Keji Zhao, et Yi Zhang. 2009. « Genome-Wide UH2A Localization Analysis Highlights Bmi1-Dependent Deposition of the Mark at Repressed Genes ». *PLOS Genetics* 5 (6): e1000506. <https://doi.org/10.1371/journal.pgen.1000506>.
- Palma-Álvarez, Raul Felipe, Constanza Daigre, Elena Ros-Cucurull, Marta Perea-Ortueta, Germán Ortega-Hernández, Ana Ríos-Landeo, Carlos Roncero, Josep Antoni Ramos-Quiroga, et Lara Grau-López. 2023. « Clinical Features and Factors Related to Lifetime Suicidal Ideation and Suicide Attempts in Patients Who Have Had Substance-Induced Psychosis across Their Lifetime ». *Psychiatry Research* 323 (mai): 115147. <https://doi.org/10.1016/j.psychres.2023.115147>.
- Pascoli, Vincent, Agnès Hiver, Ruud Van Zessen, Michaël Loureiro, Ridouane Achargui, Masaya Harada, Jérôme Flakowski, et Christian Lüscher. 2018. « Stochastic Synaptic Plasticity Underlying Compulsion in a Model of Addiction ». *Nature* 564 (7736): 366-71. <https://doi.org/10.1038/s41586-018-0789-4>.
- Pascoli, Vincent, Jean Terrier, Agnès Hiver, et Christian Lüscher. 2015. « Sufficiency of Mesolimbic Dopamine Neuron Stimulation for the Progression to Addiction ». *Neuron* 88 (5): 1054-66. <https://doi.org/10.1016/j.neuron.2015.10.017>.
- Ran, F. Ann, Patrick D. Hsu, Jason Wright, Vineeta Agarwala, David A. Scott, et Feng Zhang. 2013. « Genome Engineering Using the CRISPR-Cas9 System ». *Nature Protocols* 8 (11): 2281-2308. <https://doi.org/10.1038/nprot.2013.143>.
- Schmidl, Christian, André F. Rendeiro, Nathan C. Sheffield, et Christoph Bock. 2015. « ChIPmentation: Fast, Robust, Low-Input ChIP-Seq for Histones and Transcription Factors ». *Nature Methods* 12 (10): 963-65. <https://doi.org/10.1038/nmeth.3542>.
- Song, Allisa J., et Richard D. Palmiter. 2018. « Detecting and Avoiding Problems When Using the Cre-Lox System ». *Trends in Genetics: TIG* 34 (5): 333-40. <https://doi.org/10.1016/j.tig.2017.12.008>.
- Subramanian, Aravind, Pablo Tamayo, Vamsi K. Mootha, Sayan Mukherjee, Benjamin L. Ebert, Michael A. Gillette, Amanda Paulovich, et al. 2005. « Gene Set Enrichment Analysis: A Knowledge-Based Approach for Interpreting Genome-Wide Expression Profiles ». *Proceedings of the National Academy of Sciences* 102 (43): 15545-50. <https://doi.org/10.1073/pnas.0506580102>.
- Valjent, Emmanuel, Jesus Bertran-Gonzalez, Benjamin Aubier, Paul Greengard, Denis Hervé, et Jean-Antoine Girault. 2010. « Mechanisms of Locomotor Sensitization to Drugs of Abuse in a Two-Injection Protocol ». *Neuropsychopharmacology: Official Publication of the American College of Neuropsychopharmacology* 35 (2): 401-15. <https://doi.org/10.1038/npp.2009.143>.

Varin, Christophe, Amandine Cornil, Delphine Houtteman, Patricia Bonnavion, et Alban de Kerchove d'Exaerde. 2023. « The Respective Activation and Silencing of Striatal Direct and Indirect Pathway Neurons Support Behavior Encoding ». *Nature Communications* 14 (1): 4982. <https://doi.org/10.1038/s41467-023-40677-0>.

Vooijs, M., J. Jonkers, et A. Berns. 2001. « A Highly Efficient Ligand-Regulated Cre Recombinase Mouse Line Shows That LoxP Recombination Is Position Dependent ». *EMBO Reports* 2 (4): 292-97. <https://doi.org/10.1093/embo-reports/kve064>.

Yang, Wei, Yun-Hwa Lee, Amanda E. Jones, Jessica L. Woolnough, Dewang Zhou, Qian Dai, Qiang Wu, Keith E. Giles, Tim M. Townes, et Hengbin Wang. 2014. « The Histone H2A Deubiquitinase Usp16 Regulates Embryonic Stem Cell Gene Expression and Lineage Commitment ». *Nature Communications* 5 (mai): 3818. <https://doi.org/10.1038/ncomms4818>.

Zhang, Haiying, Douglas N. Roberts, et Bradley R. Cairns. 2005. « Genome-Wide Dynamics of Htz1, a Histone H2A Variant That Poises Repressed/Basal Promoters for Activation through Histone Loss ». *Cell* 123 (2): 219-31. <https://doi.org/10.1016/j.cell.2005.08.036>.

Zhao, Qingqing, Yang Kong, Alec Kittredge, Yao Li, Yin Shen, Yu Zhang, Stephen H Tsang, et Tingting Yang. 2021. « Distinct expression requirements and rescue strategies for BEST1 loss- and gain-of-function mutations ». Édité par Merritt Maduke, Richard W Aldrich, et Wallace B Thoreson. *eLife* 10 (juin): e67622. <https://doi.org/10.7554/eLife.67622>.

REVIEWERS' COMMENTS

Reviewer #1 (Remarks to the Author):

The authors did a great job addressing my concerns and comments from the initial review. I appreciate the analysis on distinguishing locomotor performance from cocaine-induced specific effects to understand the role of Maged1 in sensitization apart from performance issues.

The added IVSA nose poke data and CPP data add to the overall strength of the study. For the CPP behavior, please report the initial CPP scores during the pre-test. That is important to see so that one can determine whether there is an initial bias that confounds interpretation of the test. Is there a reason not to include this new behavioral data in one of the primary figures?

Many other aspects have been tightened up, clarified, and re-worded, which improves the manuscript quite a bit.

Reviewer #2 (Remarks to the Author):

Cheron et al have completed a thorough revision and comprehensive rebuttal. All of my comments have been addressed.

I do note that there are several typos and grammatical errors, including in the abstract. A careful proofread and correction is needed before publication, which should be provided by NComms staff.

The manuscript is a valuable contribution to the field and I look forward to its timely publication.

Sincerely,
Elizabeth A Heller

Reviewer #3 (Remarks to the Author):

Cheron et al responses to the comments have been both clarifying and informative. It is evident that the authors put in significant effort to improve the manuscript, rephrasing and expanding on critical sections has enhanced the overall readability and comprehension of the study both from a technical and theoretical point of view. Furthermore, I am pleased to note that the authors provided additional experimental data in response to some of the comments. In particular, the inclusion of the vGluT2-mediated Maged1 inactivation in the Cocaine Conditioned Place Preference (CPP) paradigm, and the addition of the requested imaging quantifications, strengthened the results about the effects of cocaine exposure in the context of Maged1 cKO in PVT, thus enriching the content of the manuscript, making it more informative and valuable to the reader. In its current form, the manuscript has significantly improved, and I believe it is now better suited for publication in Nature Communications.

Responses to reviewers.

Reviewer #1 (Remarks to the Author):

The authors did a great job addressing my concerns and comments from the initial review. I appreciate the analysis on distinguishing locomotor performance from cocaine-induced specific effects to understand the role of Maged1 in sensitization apart from performance issues.

We would like to thank the reviewer for his/her support.

The added IVSA nose poke data and CPP data add to the overall strength of the study. For the CPP behavior, please report the initial CPP scores during the pre-test. That is important to see so that one can determine whether there is an initial bias that confounds interpretation of the test. Is there a reason not to include this new behavioral data in one of the primary figures?

Sorry for this oversight. There was no initial bias in this experiment and have added the data of the pretest in the graph.

Many other aspects have been tightened up, clarified, and re-worded, which improves the manuscript quite a bit.

We would like to thank the reviewer for his/her support.

Reviewer #2 (Remarks to the Author):

Cheron et al have completed a thorough revision and comprehensive rebuttal. All of my comments have been addressed.

I do note that there are several typos and grammatical errors, including in the abstract. A careful proofread and correction is needed before publication, which should be provided by NComms staff.

We have carefully proof-read our manuscript (see our revised version with corrections).

The manuscript is a valuable contribution to the field and I look forward to its timely publication.

We would like to thank Elizabeth A Heller for her kind support.

Sincerely,
Elizabeth A Heller

Reviewer #3 (Remarks to the Author):

Cheron et al responses to the comments have been both clarifying and informative. It is evident that the authors put in significant effort to improve the manuscript, rephrasing and expanding on critical sections has enhanced the overall readability and comprehension of the study both from a technical and theoretical point of view. Furthermore, I am pleased to note that the authors provided additional experimental data in response to some of the comments. In particular, the inclusion of the vGluT2-mediated Maged1 inactivation in the Cocaine Conditioned Place Preference (CPP) paradigm, and the addition of the requested imaging quantifications, strengthened the results about the effects of cocaine exposure in the context of Maged1 cKO in PVT, thus enriching the content of the manuscript, making it more informative and valuable to the reader.

In its current form, the manuscript has significantly improved, and I believe it is now better suited for publication in Nature Communications.

We would like to thank the reviewer for his/her support.